# Sustained growth of sulfur hexafluoride emissions in China inferred from atmospheric observations

Minde An [1,2,3] ✉, Ronald G. Prinn [1], Luke M. Western [3,4], Xingchen Zhao[2], Bo Yao [5,6] ✉, Jianxin Hu [2], Anita L. Ganesan [1,7], Jens Mühle [8], Ray F. Weiss [8], Paul B. Krummel [9], Simon O'Doherty [3], Dickon Young [3] & Matthew Rigby [1,3]

Sulfur hexafluoride ($SF_6$) is a potent greenhouse gas. Here we use long-term atmospheric observations to determine $SF_6$ emissions from China between 2011 and 2021, which are used to evaluate the Chinese national $SF_6$ emission inventory and to better understand the global $SF_6$ budget. $SF_6$ emissions in China substantially increased from 2.6 (2.3-2.7, 68% uncertainty) Gg yr$^{-1}$ in 2011 to 5.1 (4.8-5.4) Gg yr$^{-1}$ in 2021. The increase from China is larger than the global total emissions rise, implying that it has offset falling emissions from other countries. Emissions in the less-populated western regions of China, which have potentially not been well quantified in previous measurement-based estimates, contribute significantly to the national $SF_6$ emissions, likely due to substantial power generation and transmission in that area. The $CO_2$-eq emissions of $SF_6$ in China in 2021 were 125 (117-132) million tonnes (Mt), comparable to the national total $CO_2$ emissions of several countries such as the Netherlands or Nigeria. The increasing $SF_6$ emissions offset some of the $CO_2$ reductions achieved through transitioning to renewable energy in the power industry, and might hinder progress towards achieving China's goal of carbon neutrality by 2060 if no concrete control measures are implemented.

Sulfur hexafluoride ($SF_6$) is an extremely potent greenhouse gas with a global warming potential (GWP) of ~25,000 over a 100-year time horizon[1,2]. The lifetime of $SF_6$ (~1000–3200 years[1–4]) is so long that $SF_6$ released to the atmosphere today can be considered to cause a near-permanent change to the global radiative forcing compared to the timescales of current global climate mitigation policies. Due to its substantial impact on the global climate, $SF_6$ had been incorporated into the Kyoto Protocol[5] and now into the Paris Agreement[6] under the United Nations Framework Convention of Climate Change (UNFCCC).

Emissions of $SF_6$ to the atmosphere are thought to be primarily from its use in high-voltage electrical switchgear, and, to a lesser extent, magnesium smelting and other industrial uses[7–10]. Emissions of $SF_6$ from natural sources are negligible relative to anthropogenic emissions[10–12]. Global $SF_6$ mole fractions and emissions have been

[1]Center for Global Change Science, Massachusetts Institute of Technology, Cambridge, MA 02139, USA. [2]College of Environmental Sciences and Engineering, Peking University, Beijing 100871, China. [3]School of Chemistry, University of Bristol, Bristol BS8 1TS, UK. [4]Global Monitoring Laboratory, National Oceanic and Atmospheric Administration, Boulder, CO 80305, USA. [5]Department of Atmospheric and Oceanic Sciences & Institute of Atmospheric Sciences, Fudan University, Shanghai 200438, China. [6]Meteorological Observation Centre of China Meteorological Administration (MOC/CMA), Beijing 100081, China. [7]School of Geographical Sciences, University of Bristol, Bristol BS8 1SS, UK. [8]Scripps Institution of Oceanography, University of California San Diego, La Jolla, CA 92093, USA. [9]Climate, Atmosphere and Oceans Interactions, CSIRO Environment, Aspendale, VIC 3195, Australia. ✉e-mail: mindean@mit.edu; yaobo@fudan.edu.cn

increasing rapidly since the 2000s[10,13], even though the $SF_6$ emissions reported by UNFCCC Annex-I countries have been reduced since the 1990s as a result of efforts to reduce $SF_6$ emissions in electrical equipment[9,10,14,15]. These reductions from Annex-I countries appear to be offset by the increase of $SF_6$ emissions from non-Annex-I countries (including China) due to their rapid expansion of power demand and fast adoption of renewable energy technologies[10]. The global mean annual mole fraction derived from measurements made by the Advanced Global Atmospheric Gases Experiment (AGAGE) in 2020 was more than double that in 2000[13], and the radiative forcing from $SF_6$ would increase by another factor of ~10 by 2100 if the observed growth rate of global $SF_6$ emissions over 2000–2018 continues, inferred by a previous study[9].

China is thought to be the major contributor to $SF_6$ emissions among all non-Annex-I countries due to its high electrical power demand[10]. "Bottom-up" emissions inventories, compiled based on energy and industrial activity data and emissions factors, have been reported for $SF_6$ in China previously[7,8,10,16,17]. There are officially reported $SF_6$ emissions by China (national inventories) in their national communications or biennial updates to the UNFCCC for six discrete years this century (2005[18], 2010[19], 2012[20], 2014[21], 2017[22], and 2018[23]). However, large discrepancies exist between some different bottom-up estimates. For example, the $SF_6$ emission in 2018 from the US EPA estimate[17] (1.6 Gg yr$^{-1}$) was much lower than the quantities reported by recent studies or EDGAR[8,10,16] (~4–5 Gg yr$^{-1}$), while the magnitude of the latest $SF_6$ emission submitted to the UNFCCC by China for 2018[23] (3 Gg yr$^{-1}$) falls between the estimates made by the US EPA and other studies. "Top-down" estimates, which are derived from atmospheric observations, can aid in the validation and improvement of national inventories as recommended by IPCC 2019 guidelines[24]. However, the two existing long-term time series of top-down emissions from China were both derived from atmospheric measurements made outside of China (in South Korea or Japan)[10,25], which have limited sensitivities to emissions from regions such as western China. A thorough atmospheric observation-based understanding of $SF_6$ emissions in China is currently lacking.

In this study, emissions of $SF_6$ in China over 2011–2021 were derived from atmospheric observations collected from nine sites within a Chinese measurement network and a top-down inverse modeling framework. The derived top-down emissions were compared to previous studies to evaluate national bottom-up estimates, and potential explanations for discrepancies are discussed. Substantial $SF_6$ emissions from the less-populated western regions of China were identified in our study and their potential sources are examined. Finally, the increasingly important role of China's $SF_6$ emissions in the global total emissions is discussed.

## Results

### $SF_6$ emissions in China derived from atmospheric observations

The emissions of $SF_6$ from China (defined here to be the Chinese mainland, excluding Hong Kong, Macau, and the ocean areas) over 2011–2021 are shown in Fig. 1. The atmospheric observations used to derive emissions can be found in Supplementary Fig. 1 and Supplementary Data 1. Substantial improvements in the fitness to the atmospheric observations between using the a priori and a posteriori emissions (Supplementary Tables 1, 2), and substantial uncertainty reductions (Supplementary Data 2) have been achieved during the inversion. There is a substantial growth in the derived $SF_6$ emissions in China over the period, which increased from 2.6 (2.3–2.7, 68% uncertainty, the same hereinafter) Gg yr$^{-1}$ in 2011 to 5.1 (4.8–5.4) Gg yr$^{-1}$ in 2021; i.e., by 2.6 (2.2–2.9) Gg yr$^{-1}$ or by ~100%. The magnitudes of the emissions and their increase are relatively consistent when different prior information for the emissions is used (Supplementary Discussion 1).

While the various top-down and bottom-up estimates (Fig. 1) generally show increasing $SF_6$ emissions in China, large discrepancies in magnitudes exist, especially over the study period (2011–2021). Within the uncertainties, the top-down estimates in this study are reasonably consistent in magnitude with the bottom-up EDGAR v7.0 inventory[16] (which is the a priori emissions used in the top-down inversion) and the most recently published national bottom-up estimate by Guo et al.[8]. The top-down emissions are also similar in magnitude to the bottom-up $SF_6$ emissions which were derived using data solely from the electric power industry in China, by Simmonds et al.[10] (Fig. 1a) and Zhou et al.[26] (3.5 Gg yr$^{-1}$ in 2015, which is not shown in Fig. 1). The top-down estimates in this work and these bottom-up emissions agree when different a priori emissions were used (Supplementary Discussion 1).

The top-down estimates in this study are substantially larger than the US EPA bottom-up estimate[17]. Officially reported bottom-up national emissions to the UNFCCC from China are available in six years this century, 2005[18], 2010[19], 2012[20], 2014[21], 2017[22], and 2018[23]. The first three officially reported values[18–20] are very close to the US EPA estimate[17], and substantially lower than the top-down estimates in this study. The US EPA estimate[17] and the first three officially reported values[18–20], are lower than all other top-down and bottom-up estimates in China (Fig. 1), including those that exclusively consider $SF_6$ emissions from the electric power industry[10,26]. The reason for the lower emissions in these estimates could be due to a combination of incomplete inclusion of emission source sectors, inaccuracy in activities data, and underestimation of emission factors (mainly in the electric power sector). For example, the US EPA estimate[17] does not include the $SF_6$ emissions during the manufacture of electrical equipment, which are important contributors to total $SF_6$ emissions. It is worth noting that the latest three officially reported values from China after 2014[21–23] are much closer to (although still lower than) our top-down estimate, EDGAR[16] and Guo et al.[8]. This finding may indicate that the estimation method for the Chinese national inventory has been improved between 2012 and 2014. This could be due to more accurate reporting of the quantities of $SF_6$ used in various source sectors (activity data), a more realistic representation of the process by which $SF_6$ is emitted (i.e., emissions factors), or a combination thereof. Unfortunately, no additional information is available to allow us to delve further into the reasons behind the evolutions of the compilation of individual national inventories and the differences between different bottom-up estimates.

The top-down $SF_6$ emissions for China in this study agree well with the two previous top-down estimates[10,25] during 2011–2012 (Fig. 1b), which were derived by observations made outside of China (in South Korea or Japan). However, emissions in this study are substantially larger than the only top-down estimates for the years since 2013 by Simmonds et al.[10]. Because of the limited measurements available to them, the top-down estimates in Simmonds et al.[10] only focused on eastern China, emissions from which were scaled by population to estimate the national total. A comparison of $SF_6$ emissions from eastern China between this study and Simmonds et al.[10] is illustrated in Supplementary Fig. 2, which shows that emissions in eastern China were similar between the two studies during 2011–2012, but large discrepancies emerged thereafter. The differences between the emissions for China in Simmonds et al.[10] and this study are likely to be dominated by the different sensitivities of measurements to emissions, different inverse modeling frameworks, and different prior information, combined with the influence of the assumption made to scale subregional $SF_6$ emissions to the whole of China. In addition, the top-down $SF_6$ emissions (both in this study and previous studies[10,25]) commonly exhibit some inter-annual variations during the periods, which could be informed by any changes in observations or model meteorological drivers, or could be an artifact of the model-

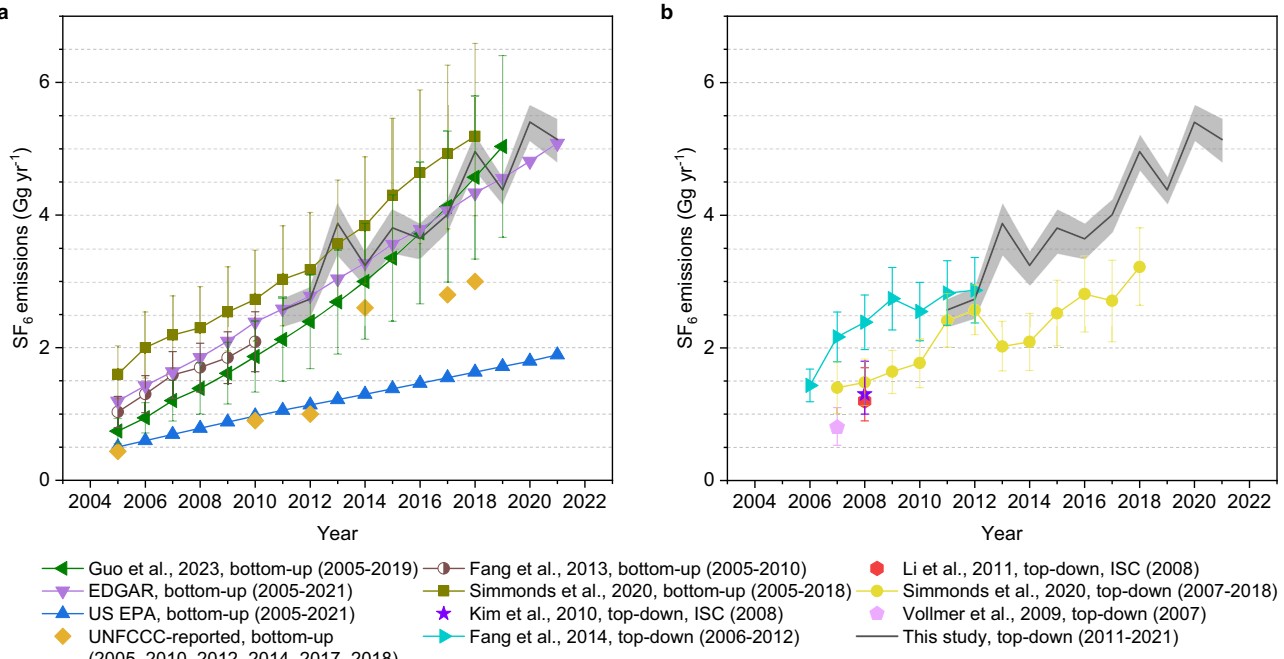

**Fig. 1 | Emissions of SF₆ in China.** Emissions of $SF_6$ in China derived in this study (black line) are compared to several previous bottom-up (plot **a**)[7,8,10,16–23] and top-down (plot **b**)[10,25,33,68,69] emission estimates. The gray shading represents the 68% uncertainty intervals of the top-down emissions in this study. Error bars for the cited emissions denote the 68% uncertainty intervals (or the 1-sigma uncertainties), with exceptions for bottom-up emissions in Fang et al.[7] where the 95% uncertainty intervals were quoted; uncertainties in the bottom-up emissions in Simmonds et al.[10] were estimated based on the range of activity data; uncertainties in the top-down emissions in Fang et al.[25] were determined through sensitivity inversion tests; and Vollmer et al.[33] defined uncertainties based on different a priori emissions. Please refer to the respective references for detailed definitions. The numbers in the parentheses after each of the legends represent the years covered by that study. All known $SF_6$ emissions in China since 2005 reported by previous studies are displayed in the plot for a complete comparison, while emissions in early years that do not overlap with the time period covered by this study are not discussed in the main text. Previous top-down estimates from Simmonds et al.[10] (yellow line in plot **b**) have focused on eastern China. They used population density as a proxy to extrapolate to a national total. The "ISC" in the legends of top-down estimates indicates the use of an interspecies correlation method in that particular study, and all other top-down studies without "ISC" in the legend used an inversion method.

measurement error, and the specific reasons to account for these variations are challenging to trace.

### Regional SF₆ emissions in China

Estimated emissions of $SF_6$ in seven subregions of China, and the contributions of each subregion to the national total increase of mean $SF_6$ emissions between 2011–2013 and 2019–2021, are shown in Fig. 2. Averaged emissions were used to calculate the emission increase, to avoid the influence from the systematic inter-annual variations in top-down results (such as due to the weaker constraint on regional emissions from the limited number of available observations in the subregion). The east of China contributes the most to total $SF_6$ emissions in China (Fig. 2a), and their increase (Fig. 2b) over the study period. This is plausible since the east of China has the most populated and industrialized areas, which are important sources of anthropogenic halogenated greenhouse gas emissions[8,10,25].

Emissions of halogenated substances outside of the east of China, including in the less-populated and developed western regions, were scarcely discussed in previous studies[27–29] due to the unavailability of measurement data within these regions. Emissions of $SF_6$ from the western regions of China were either assumed to be small[25] or were not directly quantified[10] in the two previous long-term top-down estimations. In this study, the measurements used to derive $SF_6$ emissions in China were made inside China, including from four sites within the western regions. These measurements allow us to effectively constrain the emissions in the western regions (considering the uncertainty reductions, improvements in the fitness to observations, differences between a posteriori and a priori emission spatial distributions, and

uncertainties from prior emissions, see Supplementary Discussion 2 for details). We find that the $SF_6$ emissions in Chinese regions outside of the east of China are also substantial. Emissions of $SF_6$ in the north, northwest, central, south and southwest of China contribute an average of ~18%, ~14%, ~11%, ~9%, and ~8%, respectively to the national total emissions over 2011–2021 (Supplementary Fig. 6a), and contribute substantially (10%, 7%, 12%, 14%, and 11%, respectively) to the emissions increase between 2011–2013 and 2019–2021 (Fig. 2b).

A previous bottom-up estimate of $SF_6$ emissions in China[8] showed that the power industry is the dominant source sector (which is consistent with conclusions obtained by top-down global emissions[10] and emissions in the USA[9]), followed by medical use, magnesium production, semiconductor manufacture, gas-air tracer experiments, and other minor sectors. We find that, in addition to the east of China, other regions of China, including the northwest, where the power industry[30,31] and magnesium industry[32] have intense activities (Supplementary Fig. 7), also have high emissions. These $SF_6$ emissions may be attributed to the leakage of $SF_6$ from power generation and transmission and magnesium production in these regions. The annual $SF_6$ emissions in each province in China are highly correlated with their corresponding power generation and consumption (representing the size of power industry) (Supplementary Discussion 3, Pearson correlation coefficient "$r$" = 0.83 for all years, $p < 0.01$), and are not correlated with magnesium production ("$r$" = 0.11, $p > 0.05$), indicating that the power industry may be a prominent source for $SF_6$ emissions in China, including in the western regions. The spatial patterns of power generation and consumption in China (Supplementary Fig. 7a, b) are similar to those of $SF_6$ emissions

a

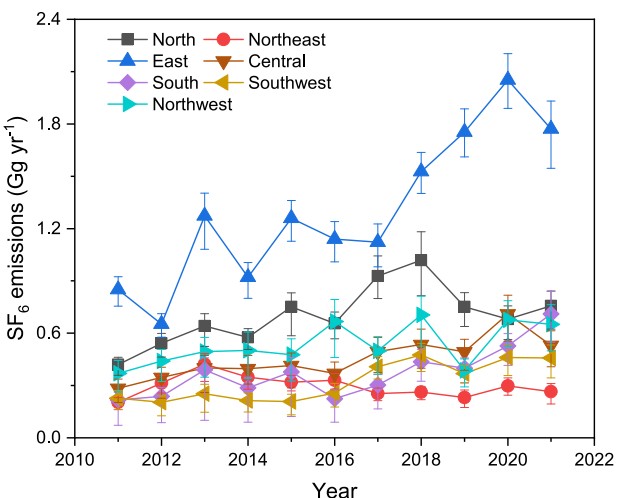

b

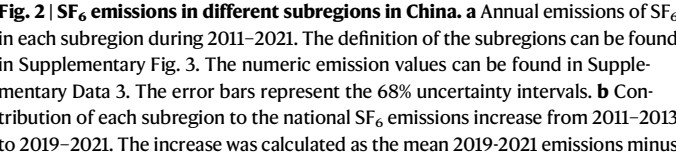

**Fig. 2 | SF$_6$ emissions in different subregions in China. a** Annual emissions of SF$_6$ in each subregion during 2011–2021. The definition of the subregions can be found in Supplementary Fig. 3. The numeric emission values can be found in Supplementary Data 3. The error bars represent the 68% uncertainty intervals. **b** Contribution of each subregion to the national SF$_6$ emissions increase from 2011–2013 to 2019–2021. The increase was calculated as the mean 2019-2021 emissions minus the mean 2011–2013 emissions. The sum of the percentages in plot **b** is larger than 100%, as the emissions from northeast of China decreased over the period. The plots for spatial distributions of SF$_6$ emissions are shown in Supplementary Fig. 4. The plots for spatial distributions of incremental emissions during the inversion (a posteriori emissions minus a priori emissions) are presented in Supplementary Fig. 5.

(Fig. 2a). In addition, the electricity supply-demand imbalance, defined as the power generation minus power consumption (Supplementary Fig. 8), underscores the inter-regional power transmission from the western regions to the eastern regions, where substantial use of SF$_6$ could lead to large emissions. These reinforce the important role of the power industry for SF$_6$ emissions in China.

Population or gross domestic product (GDP) has been used previously as a proxy for SF$_6$ emissions[8,10,33]. However, we find that the correlation between SF$_6$ emissions with either population ($r = 0.66$) or GDP ($r = 0.73$) is lower than that with power industry ($r = 0.83$) or nightlights ($r = 0.86$) (Supplementary Discussion 3), indicating that nightlights or the power industry perform as a better proxy for SF$_6$ emissions in China. The derived SF$_6$ emissions in each subregion can be different from the corresponding population or GDP as a percentage of the whole of China, especially in the northwest of China (Supplementary Fig. 6). Nevertheless, population and GDP may be more representative proxies for the other emissions source sectors of SF$_6$ in the east of China, such as the semiconductor industry and medical use, which tend to be densely located in populated and developed areas (see ref. 30,34 and Supplementary Fig. 4).

### China's contribution to global SF$_6$ emissions

Emissions of SF$_6$ in China account for an average of ~46% of global total emissions over 2011–2021, derived from trends in AGAGE global background observations (updated from Laube and Tegtmeier et al.[13] and Simmonds et al.[10], see Methods). The fraction of global emissions originating from China increased from 34% in 2011 to 57% in 2021 (Fig. 3). The increase of SF$_6$ emissions between 2011–2013 and 2019–2021 in China, 1.91 (1.69–2.16) Gg yr$^{-1}$, is nearly twice the global increase in the same period, 1.04 (0.76–1.33) Gg yr$^{-1}$. That means the increase in SF$_6$ emissions in China over the decade can not only explain all the global emission increase, but also offset -0.9 Gg yr$^{-1}$ of emission reductions elsewhere in the world.

A previous bottom–up study estimated a total decrease of ~0.8 Gg yr$^{-1}$ in SF$_6$ emissions from electrical equipment in countries other than China over 2011-2018[10], which is comparable to China's

offset to the rest-of-world reduction (-0.9, or -0.1 Gg yr$^{-1}$ using a constant prior, see Fig. 3b) derived in this study. This total decrease could be largely from Annex-I countries as a result of their regional regulations and voluntary measures to reduce SF$_6$ emissions in the power industry[9,10]. However, the total reduction in SF$_6$ emissions reported by all Annex-I countries to the UNFCCC over 2011-2021 was only -0.1 Gg yr$^{-1}$[35], which cannot explain the rest of world decline. It may otherwise imply that SF$_6$ emissions from Annex-I countries are underreported, as suggested by previous studies[9,10,36]. The overall trend in total emissions from other non-Annex-I countries apart from China remains difficult to discern due to the limited information about SF$_6$ emissions in these countries. For example, the SF$_6$ emissions from South Korea (a non-Annex-I country) have experienced a decline over 2014-2017, followed by an increase during 2017–2018, as reported in their latest national communications and biennial updates to UNFCCC, while emissions from Mexico and Brazil are increasing[37,38]. A more comprehensive understanding of SF$_6$ emissions in other non-Annex-I countries is needed, given the suggestion from a previous study that the expanding power industry and increasing emission factors in non-Annex-1 countries could contribute substantially to global SF$_6$ emissions[10].

The global emissions increase was attributed to the expansion of the power demand worldwide and especially in the developing world[10]. The substantial increases in electricity generation and consumption in China over 2011–2021 contributed ~60% of the global total increase in electricity (Supplementary Fig. 9). The percentage of electricity generation and consumption in China relative to the global total has increased substantially over 2011–2021, reaching 29% and 33% of global electricity generation and consumption in 2021, respectively (Supplementary Fig. 9). The significant role of China in the expansion of global power demand is consistent with its large contribution to the global SF$_6$ emissions increase. However, because China accounts for a larger fraction of global SF$_6$ emissions (~46%) than global power generation/consumption (~20–30%), our findings suggest a potentially higher average emission factor in China compared to the rest of the world.

a

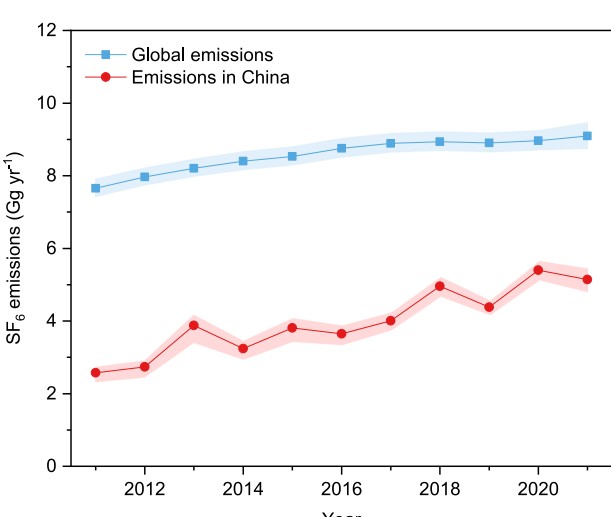

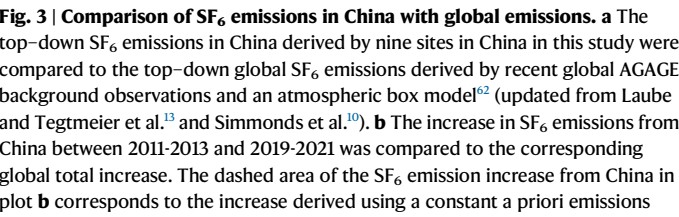

b

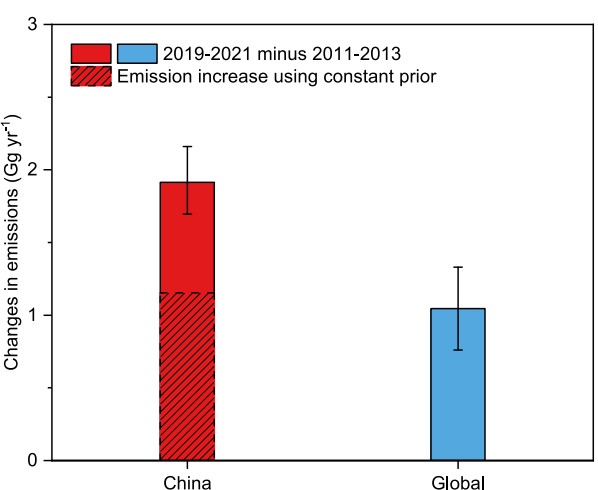

**Fig. 3 | Comparison of SF$_6$ emissions in China with global emissions. a** The top–down SF$_6$ emissions in China derived by nine sites in China in this study were compared to the top–down global SF$_6$ emissions derived by recent global AGAGE background observations and an atmospheric box model[62] (updated from Laube and Tegtmeier et al.[13] and Simmonds et al.[10]). **b** The increase in SF$_6$ emissions from China between 2011-2013 and 2019-2021 was compared to the corresponding global total increase. The dashed area of the SF$_6$ emission increase from China in plot **b** corresponds to the increase derived using a constant a priori emissions

(4 Gg yr$^{-1}$) throughout the period in the inversion (see Supplementary Discussion 1), which could exclude the potentially artificial emission increase in the a posteriori emissions resulting from the a priori emission increase itself. The dashed area represents an SF$_6$ emissions increase in China of 1.15 (0.91–1.42) Gg yr$^{-1}$, which could also be larger than the global total increase and lead to an offset of -0.1 Gg yr$^{-1}$. The shaded areas in **a** and the error bars in **b** represent the 68% uncertainty intervals or the 1-sigma uncertainties.

## Discussion

SF$_6$ is one of the most potent halogenated greenhouse gases, whose atmospheric burden is increasing rapidly[1,13]. The increase in global SF$_6$ emissions has been attributed to the rapid increase in the global power demand, especially in UNFCCC non-Annex-I (mostly developing) countries[10]. In this study, we find that China, which has dominated the expansion of the global power industry over the last decade, contributes substantially to global SF$_6$ emissions (~57% by 2021), and may have higher SF$_6$ emission factors from electricity equipment than the global average. The emissions increase from China over 2011-2021 could explain the entire global increase, and offset some decreasing emissions elsewhere. Considering that Annex-I countries are mitigating SF$_6$ emissions from the electricity industry[9,10,14,15] and the total SF$_6$ emissions from Annex-I countries may have been declining[9,10], the increasing SF$_6$ emissions in China become more important on the global scale. Thus, it is crucial to focus on SF$_6$ emissions from non-Annex-I countries, particularly China.

Top–down estimates of SF$_6$ emissions based on atmospheric observations could benefit the quality assurance of bottom–up national inventories, as recommended by the IPCC guidelines[24]. In this study, the derived top–down SF$_6$ emissions in China using nine observing sites over 2011–2021 are close to the EDGAR inventory[16] and a later bottom–up estimate by Guo et al.[8], and are substantially larger than the Chinese emissions estimated by the US EPA[17] and those reported in the Chinese national communications or biennial updates to the UNFCCC[18–23]. The discrepancy between the top–down emissions and officially reported emissions became substantially smaller in the latest reported values after 2014[21–23], suggesting an improvement in the official national inventory methodology or activity data. It is likely that more reliable industry data were used in the national inventories after 2014, and/or the actual SF$_6$ emission factor has become much closer to the emission factor used to compile the post-2014 national inventories.

Significant emissions of SF$_6$ were inferred from the sparsely populated western regions of China, probably due to the expanding power generation in those areas and the electricity transmissions from

the west to the east. We find that the size of the power industry defined by electricity consumption and/or production serves as a better indicator of provincial SF$_6$ emissions than population or GDP. Considering the expanding power industry in China, especially in the western regions, where both traditional power plants and renewable power generation facilities are growing[31], it is important to focus on SF$_6$ emissions across the country, not just in the most highly populated areas. Enhancing atmospheric measurements in the western regions could benefit the accurate quantification of SF$_6$ emissions in the regions, by conducting more densely located and higher-frequency atmospheric samplings (Supplementary Discussion 2).

The derived SF$_6$ emissions in China reached 125 (117–132) million tonnes (Mt) CO$_2$-eq yr$^{-1}$ (using GWP of 24,300 over 100 years[1]) in 2021, which is equivalent to ~1% of the national total CO$_2$ emission of China[39] and comparable to or larger than the national total CO$_2$ emissions in 2021 of the Netherlands (141 Mt yr$^{-1}$), Nigeria (137 Mt yr$^{-1}$), Belgium (96 Mt yr$^{-1}$), Qatar (96 Mt yr$^{-1}$) and Bangladesh (93 Mt yr$^{-1}$)[39]. There has been no sign of a reduction in SF$_6$ emissions derived in this study, emphasizing the enduring but uncertain importance of SF$_6$ in the future, especially in light of the ongoing and underlying reduction of other major greenhouse gases[1,13,40].

China aims to achieve carbon neutrality by 2060. Applying extensive renewable energy power generation, including photovoltaic and wind power as replacements for coal fire power plants, could be strongly beneficial to achieving this goal[41–44]. However, the annual mean increase in SF$_6$ emissions from China between 2011 and 2021, 6.5 (5.7–7.2) CO$_2$-eq Mt yr$^{-1}$ derived by linear regression, is equivalent to ~11% of the annual mean reduction in CO$_2$ emissions originating from applying renewable energy (photovoltaic and wind power) in electricity generation (see Supplementary Discussion 4 for details), and ~1% of the anticipated national CO$_2$ emissions under China's net-zero goal in 2060[40]. Photovoltaic and wind power generation is mostly located in the northwest and north of China, whose enhanced capacity in the future may exacerbate the existing inter-regional electricity supply-demand imbalance (the power generation minus power consumption) (Supplementary Fig. 8) and require new power transmission

infrastructure between the western (generation) and eastern (consumption) regions[41,42], and drive an increase in the demand for high-voltage power transmission equipment. That means continuous use of $SF_6$ in electrical equipment, if not controlled, will offset some of the benefits of applying renewable energy in power generation and might lead to uncertainties and difficulties in achieving China's carbon neutrality goal. Furthermore, considering the long lifetime of $SF_6$ (more than ~1000 years), any additional $SF_6$ emissions in this century will result in a near-permanent alteration to the global radiative budget that will persist well beyond the timeframe of the current climate policies, including China's carbon neutrality goal by 2060.

Adopting maintenance practices that minimize $SF_6$ leakage rates or using $SF_6$-free equipment or $SF_6$ substitutes, as has occurred, or been proposed, in the USA[15] and Europe[14], could help to minimize these offsets. Although there have been no specific measures to control $SF_6$ emissions in China, several proposals on $SF_6$ recycling, development of $SF_6$ substitutes, and development of better-sealed equipment have been introduced by the government since 2012, and several breakthroughs in environmentally friendly switchgear technologies have been achieved[45]. Violation of greenhouse gas emission controls has also been incorporated into criminal law according to the updated announcement from the Supreme People's Court and Supreme People's Procuratorate of China in August 2023[46]. Such controls, if widely implemented in the future, could contribute to a substantial reduction in $SF_6$ emissions and benefit mitigation of global warming and achievement of China's carbon neutrality goal by 2060. It is of vital importance to conduct continuous measurements of $SF_6$ in key source regions, including the resource-intensive western regions of China, in order to monitor $SF_6$ emissions and evaluate the efficacy of emission control regulations.

## Methods
### Atmospheric observations
The emissions of $SF_6$ in China were inferred from atmospheric observations conducted at nine sites, which are part of the China Meteorological Administration (CMA) network. The sites include the following sites: Akedala (AKD, Northwest China), Mt. Waliguan (WLG, Qinghai-Tibet Plateau, Northwest China), Longfengshan (LFS, Northeast China), Shangdianzi (SDZ, North China Plain), Jinsha (JSA, Central China), Lin'an (LAN, Yangtze River Delta region, East China), Jiangjin (JGJ, Sichuan Basin, Southwest China), Shangri-La (XGL, Southwest China) and Xinfeng (XFG, Pearl River Delta region, South China). The sites provide flask samplings (weekly, daily) or in situ (-hourly) background atmospheric measurements taken at least 10 km away from the nearest industrialized regions. Detailed information about site location and sampling frequency can be found in Supplementary Table 3. Compared to previous studies (e.g., ref. 10,25), the measurements from these sites are sensitive to emission sources across most regions of China (including the western regions), and the sensitivities to emissions do not exhibit significant inter-annual changes despite the varying measurement period of the sites (Supplementary Figs. 10 and 11).

All the flask samples were analyzed by an AGAGE 'Medusa' gas chromatographic system with a mass spectrometric detector (GC/MS)[47,48] in the CMA Beijing lab. In addition to the flask samples, there were two series of in situ measurements conducted at the SDZ site, one by an AGAGE 'Medusa' GC/MS system[47,48] every 2 hours over 2011–2012 and 2016–2021 and the other by a 2-channel gas chromatographic system with electron-capture detector (GC-ECD)[33] every 80 minutes over 2011–2020. Both the Medusa-GC/MS and GC-ECD measurements were calibrated on the SIO-05 scale[49], with the measurement of each sample bracketed by an analysis of the working standard gas. The measurement precisions were estimated at 0.98%, 0.4%, and 1% for the in situ GC-ECD, in situ Medusa-GC/MS and Medusa-GC/MS analysis of flask samples, respectively. The recovery rate of $SF_6$ from the flask

samples over 112 days was tested to be between 99.5 and 100.5%, and there were no drifts detected either in test samples or calibration standards. More detailed information about the sampling sites and sampling processes can be found in previous studies[50–52].

### Estimation of regional $SF_6$ emissions
The emissions of $SF_6$ in China were derived by a top–down inverse modeling framework, which consists of three components: atmospheric observations from the nine sites (described above), sensitivities of the atmospheric observations to emissions and boundary conditions (baselines), and a hierarchical Bayesian inference algorithm which utilizes prior information to constrain posterior results[53,54]. This framework has been described in detail by several previous studies[52,55,56]. In this subsection, we provide a brief explanation of how the sensitivities were calculated and of the hierarchical Bayesian inference algorithm.

The sensitivities of the atmospheric observations to emissions (so-called "footprints") and boundary conditions, were simulated by the UK Met Office NAME model (Numerical Atmospheric-dispersion Modeling Environment)[57], a Lagrangian particle dispersion model. The computational regional domain in this study was bounded at 5° S, 74° N and 55° E, 192° E. In the NAME model, particles were released from the sampling location within a ±10 m vertical window, at a rate of 20,000 per hour, and the model was run backwards in time for 30 days (or until the particles left the domain) prior to each measurement. The meteorological fields generated by the UK Met Office Unified Model analyses[58] were used to drive the NAME model, which have an increasing spatial resolution over the 2011-2021 period from 0.352° to 0.141° longitude and from 0.234° to 0.094° latitude, and a fixed temporal resolution of 3 h. No chemical loss was considered during the model runs for $SF_6$ since the simulation times (≤30 days) are much shorter than the $SF_6$ lifetime (-1000–3200 years[1–4]). Particle back-trajectories interacting with the surface (defined as the lowest 40 meters of the atmosphere above ground level[53]) were integrated over the 30-day period to calculate the sensitivities of the observations to the surface emissions. The NAME sensitivities to surface emissions were output in a fixed spatial resolution of 0.352° in longitude and 0.234° in latitude (grid cells). The locations of the back-trajectory particles leaving the domain were also recorded to calculate the sensitivities of the observations to the boundary conditions.

Daily flask and -hourly in situ observations, where the sum of the sensitivities to emissions from the surrounding 25 grids is more than 10% of the total sensitivity, are excluded in the calculation, to avoid potential poor performance of the transport model under stagnant conditions (e.g., Lunt et al.[59]). For the in situ data from SDZ, the Medusa-GC/MS data, considering its better precision, is preferrable to the GC-ECD data in each year, except for the years in which the Medusa-GC/MS data was not available or had poor temporal coverage (e.g. during 2013-2015 when there was a malfunction with Medusa-GC/MS system) where the GC-ECD data is used instead. All the in situ data were averaged over 24-hour time periods prior to being input into the inversion process, to reduce the influence of correlated model uncertainties over short timescales and to reduce the computational cost. The 24-hour averaging interval was chosen to be approximately consistent with the maximum sampling frequencies (-daily) from the flask samplings in this study. Either 12-hour (as has been done in a previous study[55]) or 24-hour averaging does not cause significant differences to the derived a posteriori emissions (Supplementary Fig. 12). A total of 4885 measurements were used after filtering and 24-hour averaging.

To solve for the emissions using the atmospheric observations and sensitivity (footprint) data, a hierarchical Bayesian inference algorithm, as described in detail in previous studies[53,54], was used, which utilizes prior information to constrain the posterior values. The hierarchical framework also allows the estimation of model-

observation uncertainties simultaneously during the inversion as a hyper-parameter. Three targeted parameters were primarily solved in the Bayesian inversion: emissions, boundary conditions, and uncertainties.

For emissions, the scaling factors based on their initial estimates were estimated during the inversion. The initial estimates for a priori emissions magnitudes over 2011–2021 were adopted from the annual bottom–up emissions of the EDGAR database v7.0[16], which were distributed in space by the nightlights data from NOAA Defense Meteorological Satellite Program-Operational Line-Scan System[60]. All the grid cells (described above in the computational domain) were aggregated into 150 regions based on their a priori contributions to the measurements (multiplying the a priori emission by the sensitivity value in each grid), by a quadtree algorithm[61]. As a result, regions with larger a priori contribution to the measurements (adjacent to the measurement sites or having high potential emissions) are divided into a higher spatial resolution. These 150 regions served as the fundamental units in the inversion, referred to as "basis functions" in this study, in which scaling factors for the emissions were solved. The probability distributions for the scaling factors of the a priori emissions in each basis function were assumed to be log-normal with shape parameters $\mu = 0.2$ and $\sigma = 0.8$. This prior probability distribution avoids negative emissions and constrains the posterior emissions to a reasonable magnitude during the inversion.

For the boundary conditions, the sensitivities of the observations to the boundary conditions were combined with the background mole fractions from AGAGE 12-box model inversions[13,62] (using the background mole fractions at the nearest grid in the box model output), to estimate a priori baseline mole fractions for each observation. The scaling factors for the prior background mole fractions on the four horizontal boundaries during the inversion follow a log-normal probability distribution, with shape parameters $\mu = 1$ and $\sigma = 1$.

The uncertainties in the inverse modeling consist of two parts: measurement error and model error. The measurement errors of all observations were estimated by the measurement precisions. During the averaging of the SDZ in situ data, the measurement error for the final data consists of two components: the root mean square of the measurement errors from all the data that were averaged, and the standard deviation of all the data that were averaged, which represents the variability of the measurement/atmospheric conditions during the averaging period. The model errors were estimated in the hierarchical Bayesian framework as a hyper-parameter (i.e., an uncertainty parameter explored in the inversion), which follows a uniform prior probability distribution bounded between 0 and 20 ppt. In addition to the uncertainties incorporated in the inverse modeling, different prior emission information used in the inversion could also cause differences in the posterior emission estimates, which were discussed in Supplementary Discussion 1.

To solve the hierarchical Bayesian inference framework in this study, a Markov chain Monte-Carlo (MCMC) method was used to sample the scaling factors for the emissions and boundary conditions, and the model uncertainties. The MCMC method enables one to solve the hierarchical Bayesian inference with hyper-parameter estimated, and with non-Gaussian prior probability distributions used[54] (log-normal and uniform distributions in this study). Two samplers were used to construct a $2.5 \times 10^5$ steps Markov chain: a No-U-Turn sampler (NUTS)[63] to sample the scaling factors for emissions and boundary conditions; and a slice sampler[64] to sample the hyper-parameter of model uncertainties, with the initial $5 \times 10^4$ samples from the Markov chain discarded to avoid potential biases in the early phase, or burn-in period. Prior to the formal Markov chain sampling, another $1.25 \times 10^5$ steps were sampled to tune the algorithm. The means and 68% uncertainty intervals (defined by the highest posterior density interval) of the $2 \times 10^5$ samples from the Markov chain process (after removing the "burn-in") were treated as the estimated emissions and their

uncertainties. The annual $SF_6$ emissions in China over 2011–2021 were solved separately for each year.

## Diagnosis of the regional inversion performance

To examine the performance of the inversion in estimating regional $SF_6$ emissions, several metrics were defined in this study, including the uncertainty reduction, and improvement in root mean-square-error (RMSE) and correlation between simulated and observed mole fractions.

We quantify the uncertainty reduction of the hierarchical Bayesian inversion with non-Gaussian prior probability distributions following a previous approach[65], by Eq. (1):

$$\text{UR} = 1 - \frac{\text{uncertainty}_{\text{posterior}}}{\text{uncertainty}_{\text{prior}}} \quad (1)$$

where UR represents the uncertainty reduction in a specific region, which is calculated based on posterior uncertainty (uncertainty$_{\text{posterior}}$) and prior uncertainty (uncertainty$_{\text{prior}}$). As introduced in the above section, an MCMC method was used to solve the non-Gaussian hierarchical Bayesian inference where the analytical solution for emissions and their uncertainties is not possible. The uncertainty$_{\text{posterior}}$ is obtained from the 68% highest posterior density interval from the Markov chain, while the uncertainty$_{\text{prior}}$ is the corresponding interval for prior emissions (not informed by any observations). The uncertainty reductions for emissions in China and all subregions are shown in Supplementary Data 2. It is evident from the uncertainty reductions that the measurements used in this study from the Chinese network constrain the Bayesian inversion and improve the estimates of $SF_6$ emissions in China (68% uncertainty reductions >45% in all years), as well as each subregion.

The RMSE and correlation, between the simulated and observed mole fractions, can serve as statistical metrics to evaluate the fit to the real observations by the prescribed emissions. The improvements in RMSEs obtained by a posteriori emissions compared to those by the a priori emissions are calculated by Eq. (2):

$$\text{improvement}_{\text{RMSE}} = 1 - \frac{\text{RMSE}_{\text{post}}}{\text{RMSE}_{\text{prior}}} \quad (2)$$

where RMSE$_{\text{post}}$ and RMSE$_{\text{prior}}$ are the RMSEs between the modeled and observed mole fractions calculated using a posteriori emissions and a priori emissions, respectively. The RMSE was calculated by Eq. (3):

$$\text{RMSE} = \sqrt{\frac{\sum (y_{\text{mod}} - y_{\text{meas}})^2}{N}} \quad (3)$$

where the $y_{\text{mod}}$ and $y_{\text{meas}}$ are the simulated mole fractions by the a posteriori or a priori emissions, and real observations, respectively. $N$ is the number of observations.

The improvements in correlations during the inversion (a posteriori compared to a priori) are calculated by Eq. (4):

$$\text{improvement}_{\text{corr}} = \frac{\text{corr}_{\text{post}}}{\text{corr}_{\text{prior}}} - 1. \quad (4)$$

where the corr$_{\text{post}}$ and corr$_{\text{prior}}$ are the correlations between the modeled and observed mole fractions obtained using the a posteriori emissions and a priori emissions, respectively. The correlations in this study are calculated based on the Pearson Correlation.

The improvements in RMSE and correlation are calculated for all sites in all years, which are shown in Supplementary Tables 1, 2. In most cases, the a posteriori emissions provide a better fit to the observations compared to the a priori emissions, even though the a priori emissions

that we used (EDGAR inventory distributed by nightlights) are reasonably accurate proxies for the "real emissions" (as discussed in the Results).

## Global SF$_6$ emissions estimation

Global SF$_6$ emissions over 2011–2021 used in this study were estimated using the AGAGE 12-box model[62,66] and AGAGE global background observations, which are an update to global SF$_6$ emissions previously published by Laube and Tegtmeier et al.[13] and Simmonds et al.[10]. The AGAGE background observations of SF$_6$ were made at five AGAGE global background sites, Cape Matatula, American Samoa (SMO; 14.2°S 170.6°W), Kennaook/Cape Grim, Tasmania, Australia (CGO; 40.7°S, 144.7°E), Mace Head, Ireland (MHD; 53.3°N, 9.9°W), Ragged Point, Barbados (RPB; 13.2°N, 59.4°W), and Trinidad Head, California, USA (THD; 41.1°N, 124.2°W). Detailed information about the AGAGE measurements can be found in Simmonds et al.[10]. Background mole fraction values from the five sites were assimilated in the AGAGE 12-box model to estimate the global emissions of SF$_6$. The model divides the globe into 12 boxes, bounded at 30°N, the equator, and 30°S latitudinally, and 500 hPa and 200 hPa vertically. A Bayesian framework was used, as described previously[67]. In the inversion, systematic uncertainties were considered related to transport (1%), lifetime (20%), and measurement calibration uncertainty (3%). AGAGE SF$_6$ measurements are reported on the Scripps Institution of Oceanography (SIO-05) calibration scale as parts-per-trillion (ppt) dry-air mole fractions[49].

## Data availability

Measurement data of SF$_6$ used to derive the regional emissions in China are provided in Supplementary Data 1. Measurement data of SF$_6$ from AGAGE background sites can be accessed at http://agage.mit.edu and ESS-DIVE: https://doi.org/10.15485/1909711.

## Code availability

The code and documentation for the regional inverse modeling technique "NAME-HBMCMC" used in this study are available at https://doi.org/10.5281/zenodo.10460276, which is a fork of the ACRG-Bristol v0.2.0 (https://doi.org/10.5281/zenodo.6834888). The latest version of the ACRG-Bristol repository can be found at https://github.com/ACRG-Bristol/acrg.

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

## Acknowledgements

This work was supported by the National Key Research and Development Program of China (Grant No. 2019YFC0214500) to J.H., B.Y., M.A., and X.Z., and Shanghai B&R Joint Laboratory Project (No. 22230750300) to B.Y. R.G.P., M.A., M.R., and A.L.G. were funded by NASA Grant 80NSSC21K1369 to MIT. L.M.W. received funding from the European Union's Horizon 2020 research and innovation program under Marie Skłodowska-Curie grant agreement no. 101030750. M.R. and A.L.G. also received funding from the Investigating HALocarbon Impacts on the Global Environment (InHALE) NERC Highlight Topic (NE/X00452X/1). The AGAGE calibrations, data analysis, and modeling, and the AGAGE Mace Head, Trinidad Head, Ragged Point, Cape Matatula, and Cape Grim stations, were supported in whole or in part by NASA (USA) grants 80NSSC21K1369 to MIT, and 80NSSC21K1210 and 80NSSC21K1201 to SIO. Support also came from BEIS (UK) for MaceHead, NOAA (USA) for Barbados, and CSIRO (Australia) for Cape Grim. We acknowledge the support from members of the Atmospheric Chemistry Research Group at the University of Bristol and thank the U.K. Met Office for the support and licensing for NAME.

## Author contributions

M.A. and B.Y. designed the research. M.A., supported by L.M.W., M.R., and A.L.G., conducted the regional inverse modeling. X.Z. contributed

to the analysis of industry data. B.Y. provided measurement data used in the regional inverse modeling. P.B.K., J.M., S.O'D., R.G.P., R.F.W., and D.Y. provided global measurement data and calibrations. M.R. and L.M.W. provided the AGAGE 12-box model inversion output, including global emissions and background mole fractions. M.A. led the writing of the manuscript, with contributions from R.G.P., L.M.W., M.R., A.L.G., J.M., B.Y., J.H., and all other authors.

## Competing interests

The authors declare no competing interests.
