## [Peer Review File · Nature Communications]

Sustained growth of sulfur hexafluoride emissions in China inferred from atmospheric observationsReviewer #1 (Remarks to the Author):

General comments:

1. The discussion regarding China's contribution to global SF₆ emissions raises some questions. Firstly, there is a need for a more detailed explanation in the Methods section regarding how the author obtained global SF₆ emissions data. Secondly, the author mentions that the global SF₆ emissions data is derived from Simmonds et al.¹⁰, whose top-down results for China have been consistently lower than the author's estimates since 2013, as indicated in Figure 1b. Therefore, it raises doubts about whether it is appropriate to directly compare the author's growth rates in China's SF₆ emissions with Simmonds' global SF₆ emissions growth rates. In Figure 1a, it is evident that EDGAR data aligns better with the author's estimates. Have the authors considered comparing their results with the global data from EDGAR?
2. The Methods section appears overly simplified at present. I recommend that the author provide additional details and emphasize the distinctions in methodology between this study and their previously published work. It is essential to explain the reasons for these differences. The current version appears to show no alterations, with the only distinction being the choice of species compared to the author's prior work.
3. Is there any observational evidence, whether direct SF₆ measurements or proxy species data, to corroborate the discussion on regional SF₆ contributions? Additionally, it is crucial to provide a more thorough explanation of how the high SF₆ emissions in western regions were estimated in the absence of observations. It appears that the method could be used to supplement emissions data from regions without observations. Given that the author has data on the spatial distribution of SF₆, why not consider a smaller-scale comparison like province?
4. What is the uncertainty source of the author's emission estimates?

Specific comments:

Line 52-64. This paragraph is somewhat confusing. I would recommend the author reorganize it for clarity. For instance, it might be helpful to first discuss the global trend of SF₆ emissions and then delve into the variations in SF₆ emissions trends at the national level.

Line 73. There is an extra parenthesis.

Reviewer #2 (Remarks to the Author):

This manuscript has presented a top-down estimate of SF₆ emissions in China for 2011-2021 and compared to previous studies to evaluate national bottom-up estimates. This manuscript offers some new knowledge on the SF₆ emission change in China. This study is overall well conducted and analyzed. I think the following comments shall be addressed for publication.

Specific Comments:

1. Introduction, Line 71-74:

"Large discrepancies exist between different bottom-up estimates. For example, emissions in 2018 from different inventories vary from 1.6 Gg yr⁻¹ to 5.2 Gg yr⁻¹," However, some recent bottom-up estimates were similar, for example Guo et al., 2023 and EDGAR. The largest discrepancy were selected, it was not objective. It was recommended to improve the review of bottom-up estimates.

2. Fig 1, Line 110-112:

"Previous top-down estimates from Simmonds et al. (yellow line in plot (B)) have focused on Eastern China, using population density as a proxy to extrapolate to a national total." The uncertainty of China's emissions using population density as a proxy to extrapolate to a national total might be large. Top-down estimates for eastern China from Simmonds et al. could be compared to emissions for eastern China in this study.

3. Line 118-122:

Top-down estimates in this study are reasonably consistent in magnitude with the bottom-up EDGAR v7.0 inventory, the most recently published national bottom-up estimate by Guo et al. Notably, bottom-up estimates usually increased every year. However, top-down estimates in this

study decreased in some years. The reason is worth discussing.

4. Line 122-132:

Top-down estimates in this study were usually substantially larger than the US EPA bottom-up estimate and officially reported bottom-up national. The reason is worth discussing in detail.

5. Fig 3A, Line 215:

Error bars represent uncertainty in other Figures.

6. Line 407-428:

The uncertainties of priori emissions were an important source of uncertainty in top-down emissions. The uncertainty of priori emission used in this study should be clearly described.

Reviewer #3 (Remarks to the Author):

General

The manuscript makes a valuable contribution to the field of inverse modeling, estimating SF6 emissions in China. The study improves our understanding of Chinese SF6 emissions with well-substantiated claims, is of high quality, and is generally suitable to be published. However, it requires revision.

My main critique is, that is not made clear enough how well the emissions of China (especially in western regions) can be constrained with the observations from the CMA network. The authors claim, that the western regions have not been quantified by previous studies (due to a lack of observations – to which I totally agree) but contribute significantly to the national emissions. However, when looking at Fig.S4 it seems that those significant western contributions originate from the very eastern part of the defined western regions, which actually have been identified as emission sources by previous studies (e.g. Fang et. al, 2014). I strongly recommend, providing a clearer explanation of the inversion's impact on the posterior emissions (especially in western regions). This should be achieved by quantifying the uncertainty reduction, incorporating inversion increment plots (posterior – prior), and looking at the improvement in correlation between modeled and observed mole fractions at the individual measurement sites, in order to give a complete picture.

Also, certain methodological aspects of the inversion process require further clarification. (see comments below)

Major comments:

L34: "Emissions in the less-populated western regions of China, which have not been quantified in previous measurement-based estimates, contribute significantly to the national SF6 emissions, due to substantial power generation and transmission in that area" & L163: "It has been previously assumed that SF6 emissions from western regions of China are small. However, using measurements made inside China, we are able to determine that the SF6 emissions outside of east of China, including in the less populated and developed western regions that were scarcely involved in previous studies on halogenated substances due to the unavailability of data within these regions, are also substantial."

If the authors want to make this point, I think they need to clarify how well these western regions

can be actually constrained with their inversion. What was the contribution of the western regions a priori? If I compare the posterior emissions of this study with the emissions of Fang et.al (2014), in both cases, I see significant emissions only in the very East of the defined western regions (see Fig.R1 in "Reviewer #3 Review Attachment #1"). So, I wonder if these "significant emissions" from the West are really something new, or more a matter of how regions are defined? Generally, it is important to see how much information the observations from the CMA network contain and how much the inversion impacts the posterior emissions. I think all these questions could be answered by quantifying the uncertainty reduction achieved by the inversion, incorporating inversion increment plots (posterior – prior), and showing how much the statistical parameters like the correlation between measured and observed mole fractions improve at individual measurement stations (I think Fig. S1 looks quite promising!).

Minor comments:

L34; I suggest writing "... likely due to substantial power generation" as I think this point is well discussed but not yet proven.

L37-41: "The CO₂-eq emissions of SF₆ in China in 2021 were 125 (116-38) Mt, comparable to the national total CO₂ emissions of several countries. The increasing SF₆ emissions offset some of the CO₂ reductions achieved through transitioning to renewable energy in the power industry, and will hinder progress towards achieving China's carbon neutrality goal." & L306-321

I appreciate that the authors compare the Chinese SF₆ emissions to the national total CO₂ emissions of other low-emitting countries and discuss the impact on the progress of China's carbon neutrality goal. By that, they demonstrate the impact of SF₆ emissions already in the present and the near future. However, I am concerned that the focus on the present might undermine the huge atmospheric burden of SF₆ that lies in the far future. In my view, the biggest threat of this gas is its very long atmospheric lifetime and its accumulation in the atmosphere. While the immediate impact on the current climate alone might not be substantial enough to drive action, I encourage the authors to draw more attention to the long-term impacts of SF₆ emissions in their work.

"several countries " -> I would mention these countries already at this place!

L40: "... and will hinder progress towards achieving China's carbon neutrality goal." -> might hinder

L320: "will offset some of the benefits from applying renewable energy in power generation, and lead to significant uncertainties and difficulties in achieving China's carbon neutrality goal."

I think this sentence is too strong: -> ... might lead to uncertainties and

L47: "SF₆ released to the atmosphere today contributes a near permanent change to the global radiative forcing on human timescales"

I would recommend rephrasing – I am not sure if it is clear what the authors mean and as I said, I think this is a very important point and should be given more attention!

L371: "top-down inverse modelling framework, which consists of three components: atmospheric observations from the nine sites, sensitivities of the atmospheric observations to emissions and boundary conditions, and a hierarchical Bayesian inference algorithm"

The a priori emissions are also an important component of the top-down inverse modelling framework.

L377 – 388: I think the authors need to provide some additional information at this point: What is the resolution of the UK Met Office Unified Model analysis? What is the resolution of the sensitivities? How many particles have been used? Is the optimization on a regular grid (are all the grid cells of the same size)? What's the size of the state vector?

L397: "averaged over 24-hour time periods"

Could the authors comment on the chosen interval? I would also recommend to explain the reasons for averaging in the manuscript.

L401-404: "To solve for the emissions using the atmospheric observations and sensitivity (footprint) data, a hierarchical Bayesian inference algorithm was used, where scaling factors for the emissions and boundary conditions, and the model uncertainties were sampled by the Markov Chain Monte-Carlo (MCMC) method."

At this point, the authors should write a few more sentences to explain the hierarchical Bayesian inference algorithm and the MCMC.

L407: Have the authors also considered doing seasonal inversions? I think it would be very interesting to see if the Chinese SF6 emissions have some seasonal pattern (as could be recently shown for American SF6 emissions).

L414: "Emissions inferred using different a priori assumptions were used to test the robustness of the results (Supplementary Fig. S2)." & L100: "The magnitudes of the emissions and their increase are relatively consistent when different a priori estimates for the emissions are used (Supplementary Fig. S2)."

Fang et al. (2014) identified the chosen a priori emissions as one of the biggest sources of uncertainty for East Asian SF6 emissions, so I think this point has to be discussed in more detail. Yes, the shape of the six time series shown in Fig S2 looks similar, implying some robustness of the inversion. Still, values vary by up to 1.5 Gg which is quite substantial, considering the total global emissions and the absolute posterior uncertainties. It would be important to investigate the reasons for that. I think this goes hand in hand with quantifying the inversion's impact on the posterior emissions as already recommended.

Fig.S2: I recommend adjusting the Y-axis (smaller interval but more ticks/labels). There is so much empty space and the time series are sometimes hard to distinguish. Also, fine (horizontal) grid lines would help to get a quicker overview.

Fig.1: I recommend more labels at the Y-axis and fine (horizontal) grid lines

L119: "reasonably consistent in magnitude with the bottom-up EDGAR v7.0 inventory"

I think the authors need to state here again that this has been used to define the prior. The deviation from the EDGAR v7.0 inventory will be strongly impacted by the definition of the prior emission uncertainty.

L159: "The sum of the percentages in plot (B) is larger than 100%, as the emissions from northeast of China decreased over the period"

I think it would be good to picture that in the plot (Fig. S2, B)!

L315: ... "whose enhanced capacity in the future may exacerbate the existing inter-regional electricity supply-demand imbalance ..."

Maybe a short description of what "supply-demand imbalance" means, so that the reader does not have to search for the definition in the plot.

L334: " It is of vital importance to conduct continuous measurements of SF6 in key source regions, including the resource-intensive western regions of China, in order to monitor SF6 emissions and evaluate the efficacy of emission control regulations"

I think the authors imply here, that the existing measurement network is not sufficient to cover the whole of China. As mentioned in the major comments, more information about the inversion's

impact on the posterior emissions would help to support the claim for additional observation data and might also help to identify badly covered regions.

Fig.S1: It would be good to also see the prior modeled observations and the prior modeled boundary conditions.

Fig.S9: Are these sensitivity plots somehow weighted by the frequency of the observation (averages)? Low frequent observations will have less impact on the inversions, which would be good to see in such plots. Also, I would mention that the southwest is actually not covered very well.

Table S1: The measurement altitude would also be valuable information.

References:

Fang, X., Thompson, R. L., Saito, T., Yokouchi, Y., Kim, J., Li, S., Kim, K. R., Park, S., Graziosi, F., and Stohl, A.: Sulfur hexafluoride (SF₆) emissions in East Asia determined by inverse modeling, *Atmos. Chem. Phys.*, 14, 4779–4791, <https://doi.org/10.5194/acp-14-4779-2014>,

Reviewer #3 Attachment on the following page

Fig R1. Comparison of the posterior emissions from (A) this study and from (b) Fang et.al (2014)

Response to reviewers' comments

Original manuscript number: NCOMMS-23-44240

Title: “Sustained growth of sulfur hexafluoride (SF₆) emissions in China inferred from atmospheric observations”

We thank the reviewers for their comments on the manuscript, which we have addressed in full, and have made substantial revision based on these comments. The text in italics is the reviewer's comment, followed by our response. During the revision of the manuscript, we have updated the SF₆ emission results in China. There have been slight changes to the reported quantities but no change to the conclusions drawn.

Reviewer #1 (Remarks to the Author):

General comments:

1. The discussion regarding China's contribution to global SF₆ emissions raises some questions. Firstly, there is a need for a more detailed explanation in the Methods section regarding how the author obtained global SF₆ emissions data. Secondly, the author mentions that the global SF₆ emissions data is derived from Simmonds et al.¹⁰, whose top-down results for China have been consistently lower than the author's estimates since 2013, as indicated in Figure 1b. Therefore, it raises doubts about whether it is appropriate to directly compare the author's growth rates in China's SF₆ emissions with Simmonds' global SF₆ emissions growth rates. In Figure 1a, it is evident that EDGAR data aligns better with the author's estimates. Have the authors considered comparing their results with the global data from EDGAR?

Response:

Regarding the first comment, we have added “Global SF₆ emissions estimation” to the Methods, line 581-597: “Global SF₆ emissions over 2011-2021 used in this study were estimated using the AGAGE 12-box model^{37,67} and AGAGE global background observations, which are an update to global SF₆ emissions previously published by Laube and Tegtmeier et al.¹³ and Simmonds et al.¹⁰. The AGAGE background observations of SF₆ were made at five AGAGE global background sites, Cape Matatula, American Samoa (SMO; 14.2° S 170.6° W), Kennaook/Cape Grim, Tasmania,

Australia (CGO; 40.7° S, 144.7° E), Mace Head, Ireland (MHD; 53.3° N, 9.9° W), Ragged Point, Barbados (RPB; 13.2° N, 59.4° W), and Trinidad Head, California, USA (THD; 41.1° N, 124.2° W). Detailed information about the AGAGE measurements can be found in Simmonds et al.¹⁰. Background mole fraction values from the five sites were assimilated in the AGAGE 12-box model to estimate the global emissions of SF₆. The model divides the globe into 12 boxes, bounded at 30° N, the equator and 30° S latitudinally, and 500 hPa and 200 hPa vertically. A Bayesian framework was used, as described previously⁶⁸. In the inversion, systematic uncertainties were considered related to transport (1%), lifetime (20%) and measurement calibration uncertainty (3%). AGAGE SF₆ measurements are reported on the Scripps Institution of Oceanography (SIO-05) calibration scale as parts-per-trillion (ppt) dry-air mole fractions⁵².

Regarding the second comment, we would like to clarify that the global SF₆ emissions used in this study were an update to the global emissions from Simmonds et al. and the Scientific Assessment of Ozone Depletion report, estimated using the above described AGAGE 12-box model inversion and AGAGE background measurements. The global emission estimation method is well-established and widely accepted for long-lived halogenated substances, and is independent from the regional emission estimation.

For SF₆ emissions in China, we compared our top-down emissions with different bottom-up inventories, to evaluate the emission magnitudes of these national bottom-up inventories in China, as recommended by the IPCC 2019 guidelines. The estimated SF₆ emissions in China from the EDGAR inventory show good agreement with our top-down emissions. The purpose of the comparison between SF₆ emissions in China with global emissions is to investigate China's role in global SF₆ emissions. Thus, the most widely accepted and robust global emissions that we can obtain (i.e. 12-box model top-down emissions) should be used for this purpose. We feel that comparing the top-down emissions from China in this study to the global emissions compiled by EDGAR (bottom-up inventory) does not add additional value to the paper.

2. The Methods section appears overly simplified at present. I recommend that the author provide additional details and emphasize the distinctions in methodology between this study and their previously published work. It is essential to explain the reasons for these differences. The current version appears to show no alterations, with

the only distinction being the choice of species compared to the author's prior work.

Response:

We have expanded the method for regional emissions estimation, lines 426-432: “The emissions of SF₆ in China were derived by a top-down inverse modelling framework, which consists of three components: atmospheric observations from the nine sites (described above), sensitivities of the atmospheric observations to emissions and boundary conditions (baselines), and a hierarchical Bayesian inference algorithm which utilizes prior information to constrain posterior results^{56,57}. This framework has been described in detail by several previous studies^{55,58,59}. In this subsection, we provide a brief explanation of how the sensitivities were calculated and of the hierarchical Bayesian inference algorithm”.

Additional details on NAME modeling, lines 434-450: “The sensitivities of the atmospheric observations to emissions (so called “footprints”) and boundary conditions, were simulated by the UK Met Office NAME model (Numerical Atmospheric-dispersion Modelling Environment)⁶⁰, a Lagrangian particle dispersion model. The computational regional domain in this study was bounded at 5° S, 74° N and 55° E, 192° E. In the NAME model, particles were released from the sampling location within a ±10 m vertical window, at a rate of 20,000 per hour, and the model was run backwards in time for 30 days (or until the particles left the domain) prior to each measurement. The meteorological fields generated by the UK Met Office Unified Model analyses⁶¹ were used to drive the NAME model, which have an increasing spatial resolution over the 2011-2021 period from 0.352° to 0.141° longitude and from 0.234° to 0.094° latitude, and a fixed temporal resolution of 3 h. No chemical loss was considered during the model runs for SF₆ since the simulation times (≤ 30 days) are much less than the SF₆ lifetime (~1,000-3,200 years¹⁻⁴). Particle back-trajectories interacting with the surface (defined as the lowest 40 meters of the atmosphere above ground level⁵⁶) were integrated over the 30-day period to calculate the sensitivities of the observations to the surface emissions. The NAME sensitivities to surface emissions were output in a fixed spatial resolution of 0.352° in longitude and 0.234° in latitude (grid cells)”.

Additional details on in situ data averaging, lines 461-467: “All the in situ data were averaged over 24-hour time periods prior to being input into the inversion process, to reduce the influence of correlated model uncertainties over short timescales and to reduce the computational cost. The 24-hour averaging interval was chosen to be approximately consistent with the maximum sampling frequencies (~daily) from the flask samplings in this study. Either 12-hour (as has been done in a previous study⁵⁸) or 24-hour averaging does not cause significant differences to the derived

a posteriori emissions (Supplementary Fig. S12)”.

Additional details on hierarchical Bayesian inference framework and the a priori information, lines 470-476: “To solve for the emissions using the atmospheric observations and sensitivity (footprint) data, a hierarchical Bayesian inference algorithm, as described in detail in previous studies^{56,57}, was used, which utilizes prior information to constrain the posterior values. The hierarchical framework also allows the estimation of model-observation uncertainties simultaneously during the inversion as a hyper-parameter. Three targeted parameters were primarily solved in the Bayesian inversion: emissions, boundary conditions, and uncertainties”; and lines 478-494: “For emissions, the scaling factors based on their initial estimates were estimated during the inversion. The initial estimates for a priori emissions magnitudes over 2011-2021 were adopted from the annual bottom-up emissions of the EDGAR database v7.0¹⁶, which were distributed in space by the nightlights data from NOAA Defense Meteorological Satellite Program-Operational Line-Scan System (https://ngdc.noaa.gov/eog/data/web_data/v4composites/). All the grid cells (described above in the computational domain) were aggregated into 150 regions based on their a priori contributions to the measurements (multiplying the a priori emission by the sensitivity value in each grid), by a quadtree algorithm⁶³. As a result, regions with larger a priori contribution to the measurements (adjacent to the measurement sites or having high potential emissions) are divided into a higher spatial resolution. These 150 regions served as the fundamental units in the inversion, referred to as “basis functions” in this study, in which scaling factors for the emissions were solved. The probability distributions for the scaling factors of the a priori emissions in each basis function were assumed to be log-normal with shape parameters $\mu=0.2$ and $\sigma=0.8$. This prior probability distribution avoids negative emissions and constrains the posterior emissions to a reasonable magnitude during the inversion”.

Additional details on the inversion uncertainties: please refer to the response to your major comment #4.

Additional details on the Markov chain Monte Carlo method, lines 518-532: “To solve the hierarchical Bayesian inference framework in this study, a Markov chain Monte-Carlo (MCMC) method was used to sample the scaling factors for the emissions and boundary conditions, and the model uncertainties. The MCMC method enables one to solve the hierarchical Bayesian inference with hyper-parameter estimated, and with non-Gaussian prior probability distributions used⁵⁷ (log-

normal and uniform distributions in this study). Two samplers were used to construct a 2.5×10^5 steps Markov chain: a No-U-Turn sampler (NUTS)⁶⁴ to sample the scaling factors for emissions and boundary conditions; and a slice sampler⁶⁵ to sample the hyper-parameter of model uncertainties, with the initial 5×10^4 samples from the Markov chain discarded to avoid potential biases in the early phase, or burn-in period. Prior to the formal Markov chain sampling, another 1.25×10^5 steps were sampled to tune the algorithm. The means and 68% uncertainty intervals (defined by the highest posterior density interval) of the 2×10^5 samples from the Markov chain process (after removing the “burn-in”) were treated as the estimated emissions and their uncertainties. The annual SF₆ emissions in China over 2011-2021 were solved separately for each year”.

3. Is there any observational evidence, whether direct SF₆ measurements or proxy species data, to corroborate the discussion on regional SF₆ contributions? Additionally, it is crucial to provide a more thorough explanation of how the high SF₆ emissions in western regions were estimated in the absence of observations. It appears that the method could be used to supplement emissions data from regions without observations. Given that the author has data on the spatial distribution of SF₆, why not consider a smaller-scale comparison like province?

Response:

To address the first question, we have added the relevant discussion in the caption of Supplementary Fig. S1: “We observed substantial enhancements of SF₆ atmospheric mole fractions above baseline levels at the measurement sites across different regions of China, including AKD in the northwest, LAN in the east, SDZ in the north, JGJ in the southwest and others, suggesting that significant SF₆ emissions exist throughout China”. To clarify, the reported **emission quantities and their spatial distributions (regional emissions contributions)** in this study are already an outcome of the SF₆ **concentration data (i.e., observational evidence)**, by a (well-established) regional inverse modelling approach.

To the best of our knowledge, there are no other available independent local “measurements” of SF₆ emissions or another proxy for SF₆ emissions in China to which we can compare the regional inverse emissions.

To answer the second question, we do include measurement sites in the western regions

of China in this study, including the Akedala and Mt. Waliguan in the northwest, and Shangri-La and Jiangjin in the southwest. These sites provide good sensitivity to emissions in the western regions of China (Supplementary Fig. S10-11), which allow us to determine the emissions in the western regions. In addition, some other sites we used in this study not in the western regions, such as the Shangdianzi high-frequency measurement station in northern China, also have sensitivity to emissions in the western regions (Supplementary Fig. 10-11). We have added an additional section in the Supplementary Text S2 (Supplementary lines 161-224) to discuss how the observations help us constrain the emissions in the western regions of China. In addition, we added a new plot of the changes of a posteriori emission spatial distributions after the inversion compared to the a priori spatial distributions in Supplementary Fig. S5, which reveals the changes of emissions in each region informed by the observations.

Regarding the last comment, although we do provide provincial emissions in Supplementary Data file 3, the discussion of emissions in this study are based on national and subregional scales, which can be well constrained by the regional inverse modelling using the nine sites in China (see Supplementary Data file 2 for uncertainty reductions). Emissions derived on finer resolution (i.e. provincial) could be more uncertain and impacted by the prior information, especially in regions far from measurement sites, or have low anticipated (prior) emissions (like the southwest of China).

4. *What is the uncertainty source of the author's emission estimates?*

Response:

We have revised the Methods part of the manuscript and introduced the uncertainties of the inverse modelling in lines 504-516: “The uncertainties in the inverse modelling consist of two parts: measurement error and model error. The measurement errors of all raw observations were estimated by the measurement precisions. During the averaging of the raw SDZ in situ data, the measurement error for the final data consists of two components: the root mean square of the measurement errors from all the raw data that were averaged, and the standard deviation of all the raw data that were averaged, which represents the variability of the measurement/atmospheric conditions during the averaging period. The model errors were estimated in the hierarchical

Bayesian framework as a hyper-parameter (i.e., an uncertainty parameter explored in the inversion), which follows a uniform prior probability distribution bounded between 0 and 20 ppt. In addition to the uncertainties incorporated in the inverse modelling, different prior emission information used in the inversion could also cause differences to the posterior emission estimates, which were discussed in Supplementary Text S1”.

We have also added an additional section “Uncertainties in posterior emissions using different prior emissions” to Supplementary Text S1 (Supplementary lines 99-160), which introduces the additional uncertainties from the prior emissions. Please refer to the Supplementary PDF file for details. We have discussed the uncertainties of the derived emissions from prior emissions in the main text, such as lines 107-108, lines 131-133 and Caption of Fig. 3b (lines 265-271).

Specific comments:

Line 52-64. This paragraph is somewhat confusing. I would recommend the author reorganize it for clarity. For instance, it might be helpful to first discuss the global trend of SF₆ emissions and then delve into the variations in SF₆ emissions trends at the national level.

Response:

We have revised the lines (now lines 54-62) to “Emissions of SF₆ to the atmosphere are thought to be primarily from its use in high-voltage electrical switch gear, and, to a lesser extent, magnesium smelting and other industrial uses⁷⁻¹⁰. Emissions of SF₆ from natural sources are negligible relative to anthropogenic emissions¹⁰⁻¹². Global SF₆ mole fractions and emissions have been increasing rapidly since the 2000s^{10,13}, even though the SF₆ emissions reported by UNFCCC Annex-I countries have been reduced since the 1990s as a result of efforts to reduce SF₆ emissions in electrical equipment^{9,10,14,15}. These reductions from Annex-I countries appear to be offset by the increase of SF₆ emissions from non-Annex-I countries (including China) due to their rapid expansion of power demand and fast adoption of renewable energy technologies^{10”}.

Line 73. There is an extra parenthesis.

Response:

We have removed the extra parenthesis.

Reviewer #2 (Remarks to the Author):

Specific Comments:

1. *Introduction, Line 71-74:*

“Large discrepancies exist between different bottom-up estimates. For example, emissions in 2018 from different inventories vary from 1.6 Gg yr⁻¹ to 5.2 Gg yr⁻¹,” However, some recent bottom-up estimates were similar, for example Guo et al., 2023 and EDGAR. The largest discrepancy were selected, it was not objective. It was recommended to improve the review of bottom-up estimates.

Response:

We have revised those lines (now lines 75-78) to “However, large discrepancies exist between some different bottom-up estimates. For example, emissions in 2018 from different bottom-up estimates vary between 1.6¹⁷, 3.0²³, 4.3¹⁶, 4.6⁸, and 5.2¹⁰ Gg yr⁻¹, where the difference (~3.6 Gg yr⁻¹ between the minimum and maximum values) could account for 40% of the global emissions¹³ in that year”.

2. *Fig 1, Line 110-112:*

“Previous top-down estimates from Simmonds et al. (yellow line in plot (B)) have focused on Eastern China, using population density as a proxy to extrapolate to a national total.” The uncertainty of China’s emissions using population density as a proxy to extrapolate to a national total might be large. Top-down estimates for eastern China from Simmonds et al. could be compared to emissions for eastern China in this study.

Response:

It is correct that scaling regional SF₆ emissions in eastern China to national total by population may lead to large uncertainties, as we discussed in lines 237-247 that population may not be a good proxy for SF₆ emissions in China.

Fig. 1 presents the available SF₆ emissions from the entirety of China, for a visual comparison. We have added the comparison of emissions for eastern China between this study and Simmonds et al., in Supplementary Fig. S2, and added the relevant discussion in lines 163-166: “A comparison of SF₆ emissions from eastern China between this study and Simmonds et al.¹⁰ is illustrated in Supplementary Fig. S2, which shows that emissions in eastern China were similar between the two studies during 2011-2012, but large discrepancies emerged thereafter”.

3. *Line 118-122:*

Top-down estimates in this study are reasonably consistent in magnitude with the bottom-up EDGAR v7.0 inventory, the most recently published national bottom-up estimate by Guo et al. Notably, bottom-up estimates usually increased every year. However, top-down estimates in this study decreased in some years. The reason is worth discussing.

Response:

We have added the discussion of inter-annual variations in top-down emissions in lines 170-175: “In addition, the top-down SF₆ emissions (both in this study and previous studies^{10,25}) commonly exhibit some inter-annual variations during the periods, which could be informed by any changes in observations or model meteorological drivers, or could be an artifact of the model-measurement error, and the specific reasons to account for these variations are challenging to trace”. These inter-annual variations are very common in the top-down inverse modelling studies (e.g., Rigby et al., 2019 in *Nature* and Park et al., 2021 in *Nature*), and we do not use any algorithm to smooth the top-down emissions trend in this study.

The emission magnitudes and general trends of emissions from top-down methods are more reliable than their inter-annual variations, which is the reason why we use periodical emissions to discuss emission increase. We have added the statement to lines 180-183: “Averaged emissions were used to calculate the emission increase, to avoid the influence from the systematic inter-annual variations in top-down results (such as due to the weaker constraint on regional emissions from the limited number of available observations in the subregion)”.

4. Line 122-132:

Top-down estimates in this study were usually substantially larger than the US EPA bottom-up estimate and officially reported bottom-up national. The reason is worth discussing in detail.

Response:

The top-down emissions, derived from real world observations, provide valuable evaluation of the bottom-up inventory. As we mentioned in the manuscript on lines 123-155, the top-down emissions in this study found a magnitude for national SF₆ emissions similar to the levels of the latest reported bottom-up inventory by Guo et al. and the EDGAR inventory, and indicated potential underestimates in the US EPA estimate and the first three officially reported values from the national communications or biennial updates to the UNFCCC by China.

We tried to investigate the potential reasons for these underestimates and added the relevant discussion on the lines 139-155: “The US EPA estimate¹⁷ and the first three officially reported values¹⁸⁻²⁰, are lower than all other top-down and bottom-up estimates in China (Fig. 1), including those that exclusively consider SF₆ emissions from the electric power industry^{10,29}. The reason for the lower emissions in these estimates could be due to a combination of incomplete inclusion of emission source sectors, inaccuracy in activities data, and underestimation of emission factors (mainly in the electric power sector). For example, the US EPA estimate¹⁷ does not include the SF₆ emissions during manufacture of electrical equipment, which are important contributors to total SF₆ emissions. It is worth noting that the latest three officially reported values from China after 2014²¹⁻²³ are much closer to (although still lower than) our top-down estimate, EDGAR¹⁶ and Guo et al.⁸. This finding may indicate that the estimation method for the Chinese national inventory has been improved between 2012 and 2014. This could be due to more accurate reporting of the quantities of SF₆ used in various source sectors (activity data) or a more realistic representation of the process by which SF₆ is emitted (i.e., emissions factors), or a combination thereof. Unfortunately, no additional information is available to allow us to delve further into the reasons behind the evolutions of the compilation of individual national inventories and the differences between different bottom-up estimates”.

It is challenging to determine the specific reasons behind these discrepancies, due to

the incomplete publicly available information from these bottom-up inventories. The primary findings of this study are the top-down emission magnitudes and spatial distributions, which help us investigate the emission sources of SF₆ and the potential impact of SF₆ on the global climate. Further investigation into the reasons for the discrepancies between bottom-up inventories goes beyond the scope of this study.

5. *Fig 3A, Line 215:*

Error bars represent uncertainty in other Figures.

Response:

We have added the explanation of the error bars in the Fig. 3: “The shaded areas in (A) and the error bars in (B) represent the 68% uncertainty intervals or the 1-sigma uncertainties”.

6. *Line 407-428:*

The uncertainties of priori emissions were an important source of uncertainty in top-down emissions. The uncertainty of priori emission used in this study should be clearly described.

Response:

We have added an additional section “Uncertainties in posterior emissions using different prior emissions” in Supplementary Text S1 (Supplementary lines 99-160) to discuss the uncertainties originated from the prior emissions. Different priors will slightly change the derived emissions, due to the inherence of non-Gaussian hierarchical Bayesian inference framework, but these changes are acceptable and will not change the general conclusion of this study. Please refer to the Supplementary PDF file for details.

We also discussed the uncertainties from prior emissions in the main text, lines 107-108, lines 131-133 (newly added, “The top-down estimates in this work and these bottom-up emissions agree when different a priori emissions were used (Supplementary Text S1)”, and the caption of Fig. 3 (lines 265-271).

Reviewer #3 (Remarks to the Author):

General

My main critique is, that is not made clear enough how well the emissions of China (especially in western regions) can be constrained with the observations from the CMA network. The authors claim, that the western regions have not been quantified by previous studies (due to a lack of observations – to which I totally agree) but contribute significantly to the national emissions. However, when looking at Fig. S4 it seems that those significant western contributions originate from the very eastern part of the defined western regions, which actually have been identified as emission sources by previous studies (e.g. Fang et. al, 2014). I strongly recommend, providing a clearer explanation of the inversion's impact on the posterior emissions (especially in western regions). This should be achieved by quantifying the uncertainty reduction, incorporating inversion increment plots (posterior – prior), and looking at the improvement in correlation between modeled and observed mole fractions at the individual measurement sites, in order to give a complete picture.

Also, certain methodological aspects of the inversion process require further clarification. (see comments below)

Major comments:

L34: “Emissions in the less-populated western regions of China, which have not been quantified in previous measurement-based estimates, contribute significantly to the national SF6 emissions, due to substantial power generation and transmission in that area” & L163: “It has been previously assumed that SF6 emissions from western regions of China are small. However, using measurements made inside China, we are able to determine that the SF6 emissions outside of east of China, including in the less populated and developed western regions that were scarcely involved in previous studies on halogenated substances due to the unavailability of data within these regions, are also substantial.”

If the authors want to make this point, I think they need to clarify how well these western regions can be actually constrained with their inversion. What was the contribution of the western regions a priori? If I compare the posterior emissions of this study with the emissions of Fang et.al (2014), in both cases, I see significant emissions only in the very East of the defined western regions (see Fig.R1 in "Reviewer #3 Review Attachment #1"). So, I wonder if these “significant emissions” from the West are really something new, or more a matter of how regions are defined? Generally, it is important to see how much information the observations from the CMA network contain and how

much the inversion impacts the posterior emissions. I think all these questions could be answered by quantifying the uncertainty reduction achieved by the inversion, incorporating inversion increment plots (posterior – prior), and showing how much the statistical parameters like the correlation between measured and observed mole fractions improve at individual measurement stations (I think Fig. S1 looks quite promising!).

Response:

To address the reviewer’s concerns on the constraint of SF₆ emissions in China by the observations from the CMA network, we have added three metrics to diagnose the inversion performance: uncertainty reduction, improvement in root-mean-square-error (RMSE) and improvement in correlation coefficient between simulated and observed mole fractions. The definitions of these terms are explained in an additional section in the Methods “Diagnosis of the regional inversion performance”. The values of these terms are provided in Supplementary Data file 2 for uncertainty reduction, Supplementary Table S1 for improvements in RMSE and Supplementary Table S2 for improvements in correlations. We have also added the spatial distribution plots of the emission increments during the inversion (a posteriori emissions minus a priori emissions) to Supplementary Fig. S5.

Brief discussions about the inversion performance for SF₆ emissions in China were added to lines 101-104: “Substantial improvements in the fitness to the atmospheric observations between using the a priori and a posteriori emissions (Supplementary Table S1-2), and substantial uncertainty reductions (Supplementary Data 2) have been achieved during the inversion”; lines 551-554: “It is evident from the uncertainty reductions that the measurements used in this study from the Chinese network constrain the Bayesian inversion and improve the estimates of SF₆ emissions in China (68% uncertainty reductions > 45% in all years), as well as each subregion”; and lines 576-579: “In most cases, the a posteriori emissions provide a better fit to the observations compared to the a priori emissions, even though the a priori emissions that we used (EDGAR inventory distributed by nightlights) are reasonably accurate proxies for the “real emissions” (as discussed in the Results)”.

In addition, we have added a separate section “Quantification of SF₆ emissions in the western regions” to Supplementary Text S2 (Supplementary lines 161-224), to discuss the constraint of the inversion on SF₆ emissions in the western regions of China, based on uncertainty reductions, improvements in RMSEs and correlations, the inversion increments plots (a posteriori emissions minus a priori emissions), and additional sensitivity tests of inversions done with or without the western sites. It suggests that the Chinese observation network we utilized in this study (including four sites in the western regions) provides a relatively robust constraint on SF₆ emissions in western China regions. We have also found obvious increments in the a posteriori emissions beyond the a priori emissions in the very northwest regions from the spatial distributions of the inversion increments. Please refer to the Supplementary Text S2 for details.

We have revised lines 199-210 to “Emissions of halogenated substances outside of the east of China, including in the less populated and developed western regions, were scarcely discussed in previous studies³⁰⁻³² due to the unavailability of measurement data within these regions. Emissions of SF₆ from the western regions of China were either assumed to be small²⁵ or were not directly quantified¹⁰ in the two previous long-term top-down estimations. In this study, the measurements used to derive SF₆ emissions in China were made inside China, including from four sites within the western regions. These measurements allow us to effectively constrain the emissions in the western regions (considering the uncertainty reductions, improvements in the fitness to observations, differences between a posteriori and a priori emission spatial distributions, and uncertainties from prior emissions, see Supplementary Text S2 for details). We find that the SF₆ emissions in Chinese regions outside of the east of China are also substantial”.

We acknowledge that there may still be potential limitations from the CMA network in accurately quantifying emissions in the western regions of China, as we discussed at the end of the additional paragraph “Quantification of SF₆ emissions in the western regions” of Supplementary Text S2, lines 211-216: “However, the uncertainty reductions in the northwest regions are somehow lower than in other regions, and there sometimes may not be substantial improvements in the fit to the observations after the inversion. These could be due to the limited number of available measurement sites, as well as infrequent flask sampling in these regions. Denser and more frequent sampling in the western regions would better constrain SF₆ emissions in

these crucial areas”. Thus, we have softened several statements, in Abstract line 34-37: “Emissions in the less-populated western regions of China, which have potentially not been well quantified in previous measurement-based estimates, contribute significantly to the national SF₆ emissions, likely due to substantial power generation and transmission in that area”; lines 201-203: “Emissions of SF₆ from the western regions of China were either assumed to be small²⁵ or were not directly quantified¹⁰ in the two previous long-term top-down estimations”; and lines 405-407: “Compared to previous studies (e.g. ref^{10,25}), the measurements from these sites are sensitive to emission sources across most regions of China (including the western regions)”.

Minor comments:

L34; I suggest writing “... likely due to substantial power generation ...” as I think this point is well discussed but not yet proven.

Response:

We have added “likely” to that sentence.

L37-41: “The CO₂-eq emissions of SF₆ in China in 2021 were 125 (116-38) Mt, comparable to the national total CO₂ emissions of several countries. The increasing SF₆ emissions offset some of the CO₂ reductions achieved through transitioning to renewable energy in the power industry, and will hinder progress towards achieving China’s carbon neutrality goal.” & L306-321

I appreciate that the authors compare the Chinese SF₆ emissions to the national total CO₂ emissions of other low-emitting countries and discuss the impact on the progress of China’s carbon neutrality goal. By that, they demonstrate the impact of SF₆ emissions already in the present and the near future. However, I am concerned that the focus on the present might undermine the huge atmospheric burden of SF₆ that lies in the far future. In my view, the biggest threat of this gas is its very long atmospheric lifetime and its accumulation in the atmosphere. While the immediate impact on the current climate alone might not be substantial enough to drive action, I encourage the authors to draw more attention to the long-term impacts of SF₆ emissions in their work.

“several countries “ -> I would mention these countries already at this place!

Response:

We have revised “several countries” to “several countries such as Netherlands or Nigeria”.

We enhanced the discussion regarding the long-term impacts of SF₆ on the global climate, in the Discussion lines 372-375: “Furthermore, considering the long lifetime of SF₆ (more than ~1000 years), any additional SF₆ emissions in this century will result in a near-permanent alteration to the global radiative budget that will persist well beyond the timeframe of the current climate policies including the China’s carbon neutrality goal by 2060”. We are unable to make this addition to the Abstract due to its word limit. It’s worth noting that the GWP₁₀₀-weighted emissions we used to quantify the impact from SF₆ emissions already incorporate some of the impact arising from its long lifetime.

*L40: “... and will hinder progress towards achieving China’s carbon neutrality goal.”
-> might hinder*

Response:

We have revised “will hinder” to “might hinder progress towards achieving China’s goal of carbon neutrality by 2060 if no concrete control measures are implemented”.

L320: “will offset some of the benefits from applying renewable energy in power generation, and lead to significant uncertainties and difficulties in achieving China’s carbon neutrality goal.”

I think this sentence is too strong: -> ... might lead to uncertainties and

Response:

We have revised “lead to significant uncertainties” to the “might lead to uncertainties”.

L47: “SF₆ released to the atmosphere today contributes a near permanent change to the global radiative forcing on human timescales”

I would recommend rephrasing – I am not sure if it is clear what the authors mean and as I said, I think this is a very important point and should be given more attention!

Response:

We have revised the lines (now lines 46-49) to “The lifetime of SF₆ (~1,000-3,200 years¹⁻⁴) is so long that SF₆ released to the atmosphere today can be considered to cause a near-permanent change to the global radiative forcing compared to the timescales of current global climate mitigation policies” to avoid confusion and to clarify the long-term impacts of SF₆ on the global climate.

L371: “top-down inverse modelling framework, which consists of three components: atmospheric observations from the nine sites, sensitivities of the atmospheric observations to emissions and boundary conditions, and a hierarchical Bayesian inference algorithm”

The a priori emissions are also an important component of the top-down inverse modelling framework.

Response:

We have revised that sentence to “The emissions of SF₆ in China were derived by a top-down inverse modelling framework, which consists of three components: atmospheric observations from the nine sites (described above), sensitivities of the atmospheric observations to emissions and boundary conditions (baselines), and a hierarchical Bayesian inference algorithm which utilizes prior information to constrain posterior results^{56,57}”.

We revised the lines 470-472: “To solve for the emissions using the atmospheric observations and sensitivity (footprint) data, a hierarchical Bayesian inference algorithm, as described in detail in previous studies^{56,57}, was used, which utilizes prior information to constrain the posterior values”. A detailed description of the prior information was included under the “hierarchical Bayesian inference algorithm” section.

L377 – 388: I think the authors need to provide some additional information at this point: What is the resolution of the UK Met Office Unified Model analysis? What is the resolution of the sensitivities? How many particles have been used? Is the optimization

on a regular grid (are all the grid cells of the same size)? What's the size of the state vector?

Response:

We have substantially revised the Methods for regional SF₆ emission estimation (lines 425-532) and provided more details on NAME meteorological resolution, lines 441-444: “The meteorological fields generated by the UK Met Office Unified Model analyses⁶¹ were used to drive the NAME model, which have an increasing spatial resolution over the 2011-2021 period from 0.352° to 0.141° longitude and from 0.234° to 0.094° latitude, and a fixed temporal resolution of 3 h”; NAME sensitivities resolution, lines 449-450: “The NAME sensitivities to surface emissions were output in a fixed spatial resolution of 0.352° in longitude and 0.234° in latitude (grid cells)”; particle release rate, lines 438-441: “In the NAME model, particles were released from the sampling location within a ±10 m vertical window, at a rate of 20,000 per hour, and the model was run backwards in time for 30 days (or until the particles left the domain) prior to each measurement”; and “state vectors” (basis functions in this study) and scaling, lines 483-490: “All the grid cells (described above in the computational domain) were aggregated into 150 regions based on their a priori contributions to the measurements (multiplying the a priori emission by the sensitivity value in each grid), by a quadtree algorithm⁶³. As a result, regions with larger a priori contribution to the measurements (adjacent to the measurement sites or having high potential emissions) are divided into a higher spatial resolution. These 150 regions served as the fundamental units in the inversion, referred to as “basis functions” in this study, in which scaling factors for the emissions were solved”. Please refer to the manuscript Methods section for more details.

L397: “averaged over 24-hour time periods”

Could the authors comment on the chosen interval? I would also recommend to explain the reasons for averaging in the manuscript.

Response:

We have added the explanations for the averaging and the chosen averaging interval in lines 461-467: “All the in situ data were averaged over 24-hour time periods prior to being input into the inversion process, to reduce the influence of correlated model uncertainties over short timescales and to reduce the computational cost. The 24-hour averaging interval was chosen to be

approximately consistent with the maximum sampling frequencies (~daily) from the flask samplings in this study. Either 12-hour (as has been done in a previous study⁵⁸) or 24-hour averaging does not cause significant differences to the derived a posteriori emissions (Supplementary Fig. S12)”.

L401-404: “To solve for the emissions using the atmospheric observations and sensitivity (footprint) data, a hierarchical Bayesian inference algorithm was used, where scaling factors for the emissions and boundary conditions, and the model uncertainties were sampled by the Markov Chain Monte-Carlo (MCMC) method.”

At this point, the authors should write a few more sentences to explain the hierarchical Bayesian inference algorithm and the MCMC.

Response:

We have expanded the details for Markov chain Monte-Carlo method in lines 518-532: “To solve the hierarchical Bayesian inference framework in this study, a Markov chain Monte-Carlo (MCMC) method was used to sample the scaling factors for the emissions and boundary conditions, and the model uncertainties. The MCMC method enables one to solve the hierarchical Bayesian inference with hyper-parameter estimated, and with non-Gaussian prior probability distributions used⁵⁷ (log-normal and uniform distributions in this study). Two samplers were used to construct a 2.5×10^5 steps Markov chain: a No-U-Turn sampler (NUTS)⁶⁴ to sample the scaling factors for emissions and boundary conditions; and a slice sampler⁶⁵ to sample the hyper-parameter of model uncertainties, with the initial 5×10^4 samples from the Markov chain discarded to avoid potential biases in the early phase, or burn-in period. Prior to the formal Markov chain sampling, another 1.25×10^5 steps were sampled to tune the algorithm. The means and 68% uncertainty intervals (defined by the highest posterior density interval) of the 2×10^5 samples from the Markov chain process (after removing the “burn-in”) were treated as the estimated emissions and their uncertainties. The annual SF₆ emissions in China over 2011-2021 were solved separately for each year”.

L407: Have the authors also considered doing seasonal inversions? I think it would be very interesting to see if the Chinese SF6 emissions have some seasonal pattern (as could be recently shown for American SF6 emissions).

Response:

The seasonal SF₆ emissions in China, if derived by the measurement network used in this study, could sometimes be largely impacted by the prior emission information and the number of basis functions used in the inversion, due to the less densely spatial distributed measurement sites and lower-frequency samplings from most sites compared to the American measurements used in Hu et al., especially in the early years. Thus, the seasonal SF₆ emissions in China cannot be consistently and effectively constrained for all years. Although we acknowledge the potential benefit of seasonal SF₆ emissions to the investigation of the underlying emissions sources, we are not able to accurately quantify them and do not consider them to be equally important to the annual emissions within the scope of this study.

L414: “Emissions inferred using different a priori assumptions were used to test the robustness of the results (Supplementary Fig. S2).” & L100: “The magnitudes of the emissions and their increase are relatively consistent when different a priori estimates for the emissions are used (Supplementary Fig. S2).”

Fang et al. (2014) identified the chosen a priori emissions as one of the biggest sources of uncertainty for East Asian SF₆ emissions, so I think this point has to be discussed in more detail. Yes, the shape of the six time series shown in Fig S2 looks similar, implying some robustness of the inversion. Still, values vary by up to 1.5 Gg which is quite substantial, considering the total global emissions and the absolute posterior uncertainties. It would be important to investigate the reasons for that. I think this goes hand in hand with quantifying the inversion’s impact on the posterior emissions as already recommended.

Response:

We have added a separate section “Uncertainties in posterior emissions using different prior emissions” to Supplementary Text S1 (Supplementary lines 99-160), to discuss the uncertainties originating from the prior emissions in detail and explain the reason for the relatively large uncertainties for the years 2013 and 2021. The uncertainties from the prior emissions are reasonable and understandable, and the derived emission magnitudes and their general trends are relatively robust. Please refer to the

Supplementary PDF file for details.

Fig.S2: I recommend adjusting the Y-axis (smaller interval but more ticks/labels). There is so much empty space and the time series are sometimes hard to distinguish. Also, fine (horizontal) grid lines would help to get a quicker overview.

Response:

We have revised the plot accordingly. In addition, we moved the plot to Supplementary Text S1.

Fig.1: I recommend more labels at the Y-axis and fine (horizontal) grid lines

Response:

We have revised the plot accordingly.

L119: “reasonably consistent in magnitude with the bottom-up EDGAR v7.0 inventory”

I think the authors need to state here again that this has been used to define the prior. The deviation from the EDGAR v7.0 inventory will be strongly impacted by the definition of the prior emission uncertainty.

Response:

We have revised lines 125-133 to: “Within the uncertainties, the top-down estimates in this study are reasonably consistent in magnitude with the bottom-up EDGAR v7.0 inventory (which is the a priori emissions used in the top-down inversion)¹⁶ and the most recently published national bottom-up estimate by Guo et al.⁸. The top-down emissions are also similar in magnitude to the bottom-up SF₆ emissions which were derived using data solely from the electric power industry in China, by Simmonds et al.¹⁰ (Fig. 1a) and Zhou et al.²⁹ (3.5 Gg yr⁻¹ in 2015, which is not shown in Fig. 1). The top-down estimates in this work and these bottom-up emissions agree when different a priori emissions were used (Supplementary Text S1)”.

L159: “The sum of the percentages in plot (B) is larger than 100%, as the emissions from northeast of China decreased over the period”

I think it would be good to picture that in the plot (Fig. S2, B)!

Response:

Thank you for your suggestion but we could not find an appropriate way to show this neatly in the plot.

L315: ... “whose enhanced capacity in the future may exacerbate the existing inter-regional electricity supply-demand imbalance ...”

Maybe a short description of what “supply-demand imbalance” means, so that the reader does not have to search for the definition in the plot.

Response:

A brief definition of the term “supply-demand imbalance” was given when it first appears, in lines 230-233: “In addition, the electricity supply-demand imbalance, defined as the power generation minus power consumption (Supplementary Fig. S8), underscores the inter-regional power transmission from the western regions to the eastern regions, ...”.

We have revised the lines you mentioned (now lines 364-366) to “whose enhanced capacity in the future may exacerbate the existing inter-regional electricity supply-demand imbalance (the power generation minus power consumption) (Supplementary Fig. S8)”.

L334: ” It is of vital importance to conduct continuous measurements of SF₆ in key source regions, including the resource-intensive western regions of China, in order to monitor SF₆ emissions and evaluate the efficacy of emission control regulations”

I think the authors imply here, that the existing measurement network is not sufficient to cover the whole of China. As mentioned in the major comments, more information about the inversion’s impact on the posterior emissions would help to support the claim for additional observation data and might also help to identify badly covered regions.

Response:

We have provided more information about the inversion performance in constraining SF₆ emissions in China by observation data. We also discussed the potentially insufficient aspects of the current measurement network to cover the western China.

Please refer to the response to your major comment for details. We hope that our modifications address your concerns.

The lines you mentioned (now lines 388-391) serves as a summary part of the manuscript. We have added lines 342-345: “Enhancing atmospheric measurements in the western regions could benefit the accurate quantification of SF₆ emissions in the regions, by conducting more densely located and higher-frequency atmospheric samplings (Supplementary Text S2)”.

Fig.S1: It would be good to also see the prior modeled observations and the prior modeled boundary conditions.

Response:

In the Supplementary Fig. S1, we added the residuals between the a priori simulated observations and the real observations to the right panels. It is hard to distinguish a posteriori modelled values and a priori modelled values if they are plotted together into one plot, as the a priori values we used for emissions (EDGAR spread by nightlights) and boundary conditions (AGAGE 12 box model) are already reasonably realistic.

Fig. S9: Are these sensitivity plots somehow weighted by the frequency of the observation (averages)? Low frequent observations will have less impact on the inversions, which would be good to see in such plots. Also, I would mention that the southwest is actually not covered very well.

Response:

These sensitivity plots are not weighted by the observation frequency. We have added another plot showing weighted sensitivities by the observation frequency to the Supplementary file: “Fig. S11 Sensitivities (footprints) of measurements to emissions fluxes in China. As Fig. S10 but weighted by the number of measurements from each site. These sensitivities show similar patterns to Fig. S10”.

We have added the discussion about the southwest coverage in the caption of the plot (now Fig. S10): “The measurements have a lower sensitivity to some regions in the southwest of China (mainly Xizang province), where we do not anticipate large anthropogenic SF₆ emissions.

The uncertainty reductions in the southwest of China are reasonably high (as shown in Supplementary Data 2), suggesting that the derived emissions in this region are well constrained”.

Table S1: The measurement altitude would also be valuable information.

Response:

We have added the altitude as well as the sampling height information to that table (now Supplementary Table S3).

Reviewer #1 (Remarks to the Author):

The authors have addressed my comments well.

Reviewer #2 (Remarks to the Author):

This manuscript has been improved. I recommend this paper to be published after some comments have been addressed.

Specific Comments:

1. Introduction, Lines 75-78:

"However, large discrepancies exist between some different bottom-up estimates. For example, emissions in 2018 from different bottom-up estimates vary between 1.6, 3.0, 4.3, 4.6, and 5.2 Gg yr⁻¹, where the difference (~3.6 Gg yr⁻¹ between the minimum and maximum values) could account for 40% of the global emissions in that year."

As shown in Fig. 1, most recent bottom-up estimates were similar, for example Guo et al., 2023 and EDGAR. The largest discrepancy was stressed, it was not objective. The inadequacy of individual study (e.g. US EPA) should not represent whole level of bottom-up studies. It was recommended to improve the review of bottom-up estimates.

2. Fig. 1b, Lines 117-119:

"Previous top-down estimates from Simmonds et al.¹⁰ (yellow line in plot (B)) have focused on Eastern China, using population density as a proxy to extrapolate to a national total."

As discussed in lines 237-247 that population may not be a good proxy for SF₆ emissions in China. Therefore, the uncertainty of China's emissions using population density as a proxy to extrapolate to a national total might be large. The national total of China from Simmonds et al may not be shown in Fig. 1b. The results of Fig. S2 "SF₆ emissions in eastern China derived in this study compared to Simmonds et al." could be shown in Fig. 1.

Reviewer #3 (Remarks to the Author):

The authors have effectively addressed my concerns. Much effort was made to illustrate the inversion's impact on posterior emissions and to clarify methodological aspects of the inversion process. The manuscript now provides a clear and comprehensive picture, and I extend my congratulations to the authors for their interesting and well-presented work!

Response to reviewers' comments

Original manuscript number: NCOMMS-23-44240(A)

Title: "Sustained growth of sulfur hexafluoride emissions in China inferred from atmospheric observations"

Reviewer #2 (Remarks to the Author):

Specific Comments:

1. Introduction, Lines 75-78:

"However, large discrepancies exist between some different bottom-up estimates. For example, emissions in 2018 from different bottom-up estimates vary between 1.6, 3.0, 4.3, 4.6, and 5.2 Gg yr⁻¹, where the difference (~3.6 Gg yr⁻¹ between the minimum and maximum values) could account for 40% of the global emissions in that year."

As shown in Fig. 1, most recent bottom-up estimates were similar, for example Guo et al., 2023 and EDGAR. The largest discrepancy was stressed, it was not objective. The inadequacy of individual study (e.g. US EPA) should not represent whole level of bottom-up studies. It was recommended to improve the review of bottom-up estimates.

Response:

Thanks for the suggestion. We have revised the lines to: "For example, the SF₆ emission in 2018 from the US EPA estimate¹⁷ (1.6 Gg yr⁻¹) was much lower than the quantities reported by recent studies or EDGAR^{8,10,16} (~4-5 Gg yr⁻¹), while the magnitude of the latest SF₆ emission submitted to the UNFCCC by China for 2018²³ (3 Gg yr⁻¹) falls between the estimates made by the US EPA and other studies."

Before we utilize the top-down emissions derived in this study to validate these previous emissions (in the Results section), we lack evidence (in the Introduction section) to determine which inventories (whether US EPA or EDGAR, etc.) are more representative than others.

2. Fig. 1b, Lines 117-119:

"Previous top-down estimates from Simmonds et al.¹⁰ (yellow line in plot (B)) have focused on Eastern China, using population density as a proxy to extrapolate to a national total."

As discussed in lines 237-247 that population may not be a good proxy for SF₆ emissions in China. Therefore, the uncertainty of China's emissions using population density as a proxy to extrapolate to a national total might be large. The national total of China from Simmonds et al may not be shown in Fig. 1b. The results of Fig. S2 "SF₆ emissions in eastern China derived in this study compared to Simmonds et al." could be shown in Fig. 1.

Response:

Thanks for the advice. However, we believe it is better to keep Fig. 1 as it is.

To clarify, the SF₆ emissions in China cited from Simmonds et al. are the quantities directly reported by that study, which were calculated by scaling eastern China emissions to national total by Simmonds et al. (not by this study). We put the previously reported numbers in Fig. 1, to provide a comprehensive overview of SF₆ emissions in the whole of China (rather than a specific region). Basic information about these previous emissions (e.g., estimation method and assumptions made) was given in the caption of Fig. 1. Further discussions regarding these previous SF₆ emissions and their associated uncertainties are presented in subsequent paragraphs in this paper. We feel that adding emissions from eastern China to Fig. 1 disrupts the manuscript's logical flow.

We have revised the lines in the caption of Fig.1 to avoid confusion: "All known SF₆ emissions in China since 2005 reported by previous studies are displayed in the plot for a complete comparison ... Previous top-down estimates from Simmonds et al.¹⁰ (yellow line in plot **(b)**) have focused on Eastern China. They used population density as a proxy to extrapolate to a national total."